# Mutant ASXL1 induces age-related expansion of phenotypic hematopoietic stem cells through activation of Akt/mTOR pathway

Takeshi Fujino[1], Susumu Goyama[1], Yuki Sugiura[2], Daichi Inoue[3,4], Shuhei Asada [1,5], Satoshi Yamasaki[6], Akiko Matsumoto[6], Kiyoshi Yamaguchi[7], Yumiko Isobe[7], Akiho Tsuchiya[1], Shiori Shikata[1], Naru Sato[1], Hironobu Morinaga[8], Tomofusa Fukuyama [1], Yosuke Tanaka[1], Tsuyoshi Fukushima[1], Reina Takeda[1], Keita Yamamoto[1], Hiroaki Honda[5], Emi K. Nishimura [8], Yoichi Furukawa[7], Tatsuhiro Shibata[6], Omar Abdel-Wahab [3], Makoto Suematsu [2] & Toshio Kitamura [1✉]

Somatic mutations of *ASXL1* are frequently detected in age-related clonal hematopoiesis (CH). However, how *ASXL1* mutations drive CH remains elusive. Using knockin (KI) mice expressing a C-terminally truncated form of ASXL1-mutant (ASXL1-MT), we examined the influence of ASXL1-MT on physiological aging in hematopoietic stem cells (HSCs). HSCs expressing ASXL1-MT display competitive disadvantage after transplantation. Nevertheless, in genetic mosaic mouse model, they acquire clonal advantage during aging, recapitulating CH in humans. Mechanistically, ASXL1-MT cooperates with BAP1 to deubiquitinate and activate AKT. Overactive Akt/mTOR signaling induced by ASXL1-MT results in aberrant proliferation and dysfunction of HSCs associated with age-related accumulation of DNA damage. Treatment with an mTOR inhibitor rapamycin ameliorates aberrant expansion of the HSC compartment as well as dysregulated hematopoiesis in aged ASXL1-MT KI mice. Our findings suggest that ASXL1-MT provokes dysfunction of HSCs, whereas it confers clonal advantage on HSCs over time, leading to the development of CH.

[1] Division of Cellular Therapy, The Institute of Medical Science, The University of Tokyo, Minato-ku, Tokyo, Japan. [2] Department of Biochemistry, Keio University School of Medicine, and Japan Science and Technology Agency (JST), Exploratory Research for Advanced Technology (ERATO), Suematsu Gas Biology Project, Shinjuku-ku, Tokyo, Japan. [3] Human Oncology and Pathogenesis Program, Memorial Sloan—Kettering Cancer Center and Weill Cornell Medical College, New York, USA. [4] Department of Hematology-Oncology, Institute of Biomedical Research and Innovation, Foundation for Biomedical Research and Innovation at Kobe, Kobe City, Hyogo, Japan. [5] Field of Human Disease Models, Major in Advanced Life Sciences and Medicine, Tokyo Women's Medical University, Shinjuku-ku, Tokyo, Japan. [6] Laboratory of Molecular Medicine, Human Genome Center, The Institute of Medical Science, The University of Tokyo, Minato-ku, Tokyo, Japan. [7] Division of Clinical Genome Research, Advanced Clinical Research Center, The Institute of Medical Science, The University of Tokyo, Minato-ku, Tokyo, Japan. [8] Department of Stem Cell Biology, Medical Research Institute, Tokyo Medical and Dental University, Bunkyo-ku, Tokyo, Japan. ✉email: kitamura@ims.u-tokyo.ac.jp

Billions of blood cells are produced daily to maintain homeostasis in a human body. Hematopoietic stem cells (HSCs) are a rare population, but they are capable of self-renewal and multi-lineage differentiation, which play a pivotal role in maintaining life-long hematopoiesis. In the aging process, HSCs are subject to both cell-intrinsic and cell-extrinsic stresses, which may result in cellular senescence and increased incidence of malignancy[1–3]. It has been reported that aging is associated with a significant expansion in surface marker-defined immunophenotypic HSCs (pHSCs), while these cells reduce repopulation ability[4–7]. Progenitors with limited differentiation potential and impaired repopulation ability accumulate within the pHSC compartment, which may serve as the mechanistic basis for this phenomenon[5–9]. Such pHSCs may also contribute to the development of clonal hematopoiesis (CH) or clonal hematopoietic disease, but this concept has yet to be validated[1,9].

*ASXL1*, a gene frequently mutated in CH, is one of three human homologs of the *Drosophila Asx* gene and is involved in epigenetic regulation[10–13]. In addition to CH, somatic mutations of *ASXL1* gene are detected in myeloid neoplasms including myelodysplastic syndromes (MDS), chronic myelomonocytic leukemia (CMML), and acute myeloid leukemia (AML)[14–18]. Most *ASXL1* mutations detected in CH and hematological malignancies are frameshift or nonsense mutations in the last exon, generating a C-terminally truncated form of ASXL1[14,15,17–21]. Such truncated ASXL1 proteins are indeed expressed in leukemic cells and likely confer change-of-function[22]. Recently, we and others have shown that the mutant ASXL1 interacts with a deubiquitinase BAP1 to form the Polycomb-repressive deubiquitinase (PR-DUB) complex, which efficiently deubiquitinates H2AK119Ub[23,24]. The hyperactive PR-DUB complex of mutant ASXL1 and BAP1 upregulates several target genes, including *HOX* genes, by deubiquitinating H2AK119Ub to promote myeloid leukemogenesis[23]. In hematopoietic lineage-specific conditional knockin (KI) mice expressing a C-terminally truncated form of mutant ASXL1 (ASXL1-MT; 1900−1922del;E635RfsX15)[25], global reduction was observed in H3K4me3 and H2AK119Ub. In particular, levels of H3K4me3 at the loci of erythroid differentiation-related genes, such as *Id3* and *Sox6*, were markedly decreased, suggesting that ASXL1-MT impairs hematopoiesis through dysregulated epigenetic modifications.

It is now generally recognized that cells that are well-adapted to selective pressures acquire a fitness advantage, leading to clonal expansion during aging[26–29]. Clonal expansion of such blood cells harboring somatic mutations is termed CH. CH is commonly observed in elderly individuals in the absence of apparent hematological abnormalities[19–21]. Despite normal hematological parameters, individuals with CH are at increased risk of hematological malignancies, indicating that CH can be a preleukemic condition. Mutations most commonly detected in CH include the epigenetic regulators *DNMT3A*, *TET2*, and *ASXL1*. From the results of transplantation experiments using *Tet2*-deficient or *Dnmt3a*-deficient mice, the increased self-renewal of HSCs could explain why loss-of-function mutations of *DNMT3A* and *TET2* are frequently detected in CH[30–33]. On the other hand, several recent studies have shown reduced numbers and functions of HSCs in mutant ASXL1 KI mice[25,34,35]. Thus, identifying how ASXL1 mutations promote the development of CH has remained an unsolved issue.

In the present study, we examined the influence of ASXL1-MT on physiological aging using ASXL1-MT KI mice. We found that ASXL1-MT confers a competitive disadvantage on HSCs after transplantation. On the other hand, in genetic mosaic mouse model, we observed stronger growth advantage in LT-HSCs expressing ASXL1-MT specifically in native hematopoiesis, recapitulating CH in humans. As for the molecular mechanisms by which ASXL1-MT induces CH, we show that ASXL1-MT binds and activates Akt by stabilizing phosphorylated Akt in concert with Bap1. Activation of Akt/mTOR pathway induced by ASXL1-MT causes aberrant cell cycle progression and proliferation in the HSC compartment. At the same time, it also provokes dysfunction of HSCs associated with mitochondrial activation, elevated ROS levels, and increased DNA damage. These molecular changes can cause CH with increased risk of leukemogenesis, which can be ameliorated by rapamycin treatment.

## Results

**ASXL1-MT reduces the number and function of HSPCs in young mice.** To study the effects of ASXL1-MT on hematopoiesis, we crossbred ASXL1-MT KI mice with *Vav-Cre* transgenic mice. Consistent with our previous reports[25], young *Vav-Cre* ASXL1-MT KI mice did not show significant changes in hematological parameters (hereinafter, all experiments were performed with 6–12-week-old for young mice) (Fig. 1a). We also observed no significant changes in the frequency of myeloid cells (CD11b+), B cells (B220+), and T cells (CD3+) between control and *Vav-Cre* ASXL1-MT KI mice (Fig. 1b). Young *Vav-Cre* ASXL1-MT KI mice showed normal bone marrow cellularity (Fig. 1c), but exhibited a significant decrease in the frequency of hematopoietic stem and progenitor cells (HSPCs), including Lin−Sca1+c-kit+ (LSK) cells, multipotent progenitors (MPPs; CD48+CD150− LSK) and long-term HSCs (LT-HSCs; CD48−CD150+ LSK) (Fig. 1d and Supplementary Fig. 1). Competitive transplantation assays using whole bone marrow cells or defined populations (MPPs and LT-HSCs) revealed impaired repopulation ability and skewed differentiation potential toward myeloid lineage in ASXL1-MT KI HSPCs (Fig. 1e, f, Supplementary Fig. 2a–d). Thus, consistent with the previous reports, ASXL1-MT reduced the number and function of HSPCs in young mice.

To characterize the defective HSPC function of young *Vav-Cre* ASXL1-MT KI mice, we evaluated apoptosis and cell cycle status. The frequency of Annexin V-positive cells was significantly increased in HSPCs from *Vav-Cre* ASXL1-MT KI mice (Fig. 1g), indicating that ASXL1-MT KI HSPCs are more prone to apoptosis than are normal HSPCs. Ki-67/DAPI staining revealed a significantly lower frequency of LT-HSCs in G0 phase, and a higher frequency of both MPPs and LT-HSCs in S/G2/M phase in *Vav-Cre* ASXL1-MT KI mice (Fig. 1h). These data suggest that increased apoptosis and loss of quiescence contribute to the defective repopulation ability in HSPCs of young *Vav-Cre* ASXL1-MT KI mice.

**ASXL1-MT confers a clonal advantage on LT-HSCs specifically in native hematopoiesis.** Next, we assessed age-associated changes in hematopoiesis driven by ASXL1-MT. Analyses of peripheral blood cells showed mild leukocytopenia and mild anemia, which are prominent when compared with age-matched control mice, as well as moderate thrombocytosis in aged *Vav-Cre* ASXL1-MT KI mice (hereinafter, all experiments were performed with 20–24-month-old for aged mice) (Fig. 2a). Aged *Vav-Cre* ASXL1-MT KI mice exhibited a myeloid-biased hematopoiesis and hypocellular bone marrow, suggesting dysfunction of hematopoiesis (Figs. 2b, 9d, and e). Aged *Vav-Cre* ASXL1-MT KI mice do not develop apparent hematological diseases and their lifetime is equal to wild-type mice. Interestingly, the frequencies of LT-HSCs were markedly increased in aged *Vav-Cre* ASXL1-MT KI mice (Fig. 2c). We then assessed the functions of LT-HSCs in aged *Vav-Cre* ASXL1-MT KI mice. Single-cell liquid cultures revealed a decrease in the frequency of colony-forming LT-HSC from aged *Vav-Cre* ASXL1-MT KI mice compared with age-matched control mice (Supplementary Fig. 3a and b).

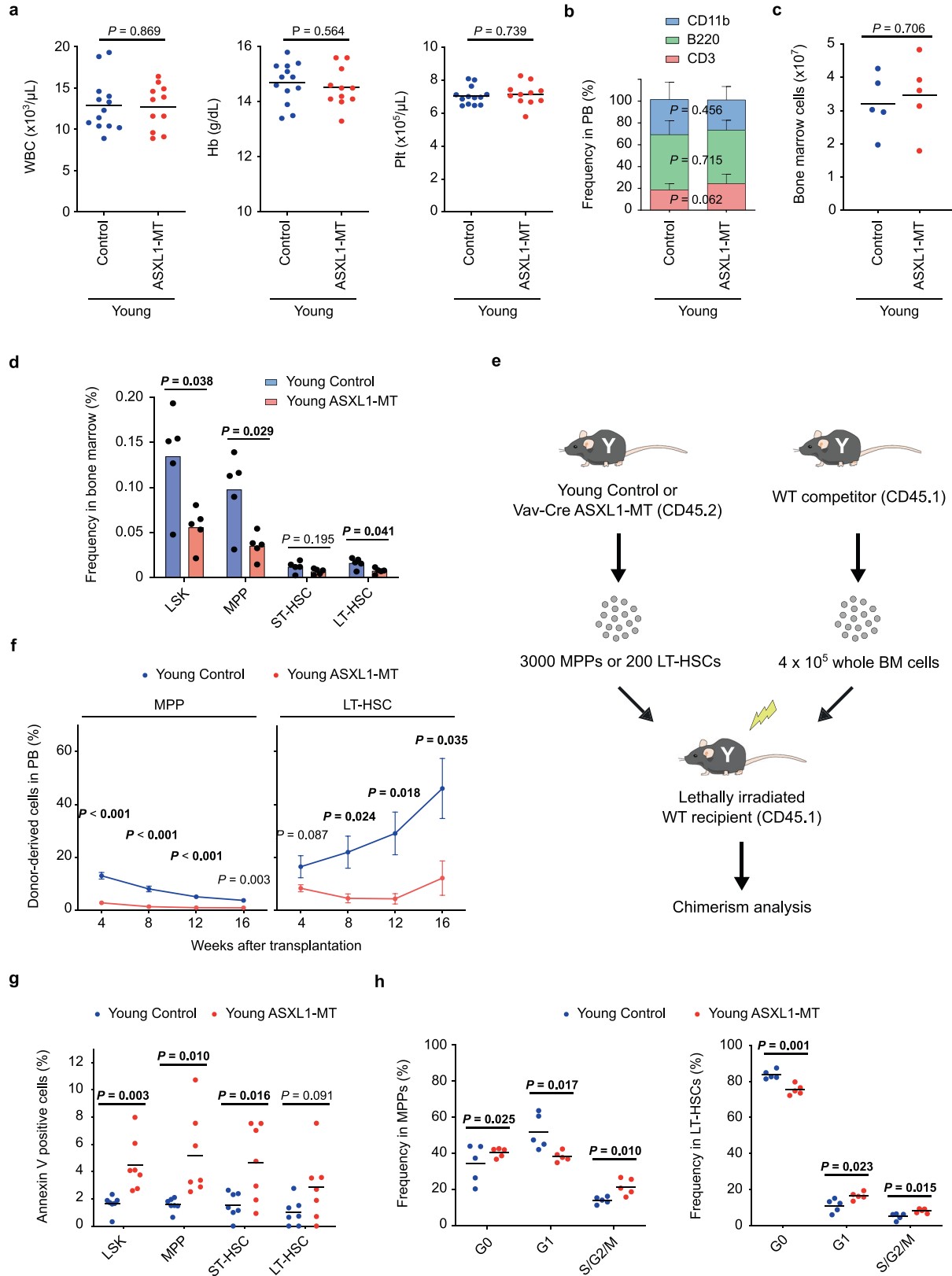

Furthermore, competitive transplantation assays using LT-HSCs from aged *Vav-Cre* ASXL1-MT KI mice and age-matched control mice showed that aged LT-HSCs expressing ASXL1-MT exhibited markedly compromised production of peripheral blood and tended to exhibit impaired regeneration of bone marrow cells including LT-HSCs (Fig. 2d–f). We also performed homing assays using LSK cells from both young and aged *Vav-Cre* ASXL1-MT KI mice. These analyses revealed that LSK cells from aged *Vav-Cre* ASXL1-MT KI mice exhibited a significant decrease in homing capability compared to age-matched control mice in addition to intrinsic defects detected by in vitro colony forming assays (Supplementary Fig. 3a–d). Thus, the increased LT-HSCs

**Fig. 1 ASXL1-MT causes dysfunction of HSPCs associated with increased apoptosis and altered cell cycle status. a** Enumeration of white blood cells (WBC), hemoglobin (Hb), and platelets (Plt) in peripheral blood of young *Vav-Cre* ASXL1-MT KI mice ($n = 13$ (Control), 11 (ASXL1-MT)). **b** Frequency of myeloid cells (CD11b$^+$), B cells (B220$^+$), and T cells (CD3$^+$) in peripheral white blood cells of young *Vav-Cre* ASXL1-MT KI mice ($n = 13$ (Control), 11 (ASXL1-MT)). **c** Absolute numbers of bone marrow cells per leg in young *Vav-Cre* ASXL1-MT KI mice ($n = 5$). **d** Frequency of LSK cells, multipotent progenitors (MPPs), short-term HSCs (ST-HSCs) and long-term HSCs (LT-HSCs) in bone marrow cells of young *Vav-Cre* ASXL1-MT KI mice ($n = 5$). **e** The experimental design for competitive transplantation assays. 3000 MPPs or 200 LT-HSCs isolated from young control or young *Vav-Cre* ASXL1-MT KI mice were transplanted into lethally irradiated recipient mice with $4 \times 10^5$ whole bone marrow cells. **f** Levels of donor chimerism in peripheral blood were analyzed at the indicated weeks after transplantation ($n = 3$ (Control), 4 (ASXL1-MT)). Data are mean ± s.e.m. **g** Apoptosis analysis of HSPCs of young *Vav-Cre* ASXL1-MT KI mice ($n = 7$). **h** Cell cycle analysis with Ki-67/DAPI staining of MPPs (left panel) and LT-HSCs (right panel) of young *Vav-Cre* ASXL1-MT KI mice ($n = 5$). Data are mean ± s.d. unless otherwise noted. $^*P \leq 0.05$, $^{**}P \leq 0.01$, $^{***}P \leq 0.001$; two-tailed Student's *t*-test.

in aged *Vav-Cre* ASXL1-MT KI mice are not functional HSCs with long-term repopulation potential (hereinafter, referred to as "immunophenotypic LT-HSCs (pLT-HSCs)").

It appears that ASXL1-MT promotes age-related changes in the hematopoietic system of *Vav-Cre* ASXL1-MT KI mice (e.g. anemia, myeloid-skewed differentiation, and hypocellular bone marrow). Therefore, we next evaluated the frequency of CD41-positive pLT-HSCs, which has been reported to skew differentiation toward myeloid and megakaryocytic lineages, and to expand during aging within the pLT-HSC compartment[8,9,36,37]. This analysis revealed that most pLT-HSCs were positive for CD41 in aged *Vav-Cre* ASXL1-MT KI mice, while 50–70% were positive in age-matched control mice (Supplementary Fig. 3e). These data imply that ASXL1-MT accelerates the physiological aging of LT-HSCs.

Next, to assess the competitive growth of ASXL1-MT KI cells in native hematopoiesis without transplantation, we bred ASXL1-MT KI mice with *Mx1-Cre* transgenic mice and partially induced expression of ASXL1-MT by polyinosine–polycytidine (pIpC) injections (Fig. 3a). As these mice carry the floxed allele of ASXL1-MT-IRES-GFP, cells expressing ASXL1-MT also express GFP (Supplementary Fig. 1). In these genetic mosaic mice, frequency of ASXL1-MT expressing cells (GFP positive) gradually increased in peripheral blood (Fig. 3b). Similar to the phenotypes of aged *Vav-Cre* ASXL1-MT KI mice, expression of ASXL1-MT in this model also caused anemia, thrombocytosis, and myeloid-biased differentiation with age (Fig. 3c and Supplementary Fig. 4a–d). The frequency of LT-HSCs expressing ASXL1-MT was increased in aged *Mx1-Cre* ASXL1-MT KI mice along with progression of cell cycle (Fig. 3d and e). Eventually, LT-HSCs expressing ASXL1-MT occupied bone marrow to outcompete their normal counterpart 2 years after pIpC injections, recapitulating CH in humans (Fig. 3f). Given that levels of pro-inflammatory cytokines including TNF-α and IFN-γ were similar in aged control and aged *Mx1-Cre* ASXL1-MT KI mice (Supplementary Fig. 4e and f), these phenotypic changes were caused by ASXL1-MT itself, not by the inflammation-induced *Mx1-Cre* expression.

Thus, ASXL1-MT causes a competitive disadvantage of LT-HSCs during transplantation; nevertheless, it confers a clonal advantage on LT-HSCs in native hematopoiesis. To investigate cell intrinsic and extrinsic mechanisms that mediate the effect of ASXL1-MT on HSC function, we performed transplantation assays with three different settings (Supplementary Fig. 5a, d, and f). First, we transplanted bone marrow cells from young *Mx1-Cre* ASXL1-MT KI mice into wild-type recipient mice and induced ASXL1-MT expression by pIpC injections one month after transplantation. Induction of ASXL1-MT after engraftment did not lower chimerism of peripheral blood but rather tended to increase the frequency of donor-derived bone marrow cells, suggesting that ASXL1-MT exerts competitive disadvantage on LT-HSCs, specifically during recovery from transplantation (Supplementary Fig. 5a–c). Second, we transplanted bone marrow cells from young *Vav-Cre* ASXL1-MT KI mice into young or aged

wild-type recipient mice. The frequencies of ASXL1-MT-expressing cells were similar regardless of the age of recipient mice (Supplementary Fig. 5d and e). Finally, we transplanted bone marrow cells from young *Vav-Cre* ASXL1-MT KI mice into *Mx1-Cre* ASXL1-MT KI mice that had been induced expression of ASXL1-MT by pIpC injections one month before transplantation. Expression of ASXL1-MT in non-hematopoietic cells of the recipient bone marrow did not affect the chimerism of ASXL1-MT-expressing cells in peripheral blood (Supplementary Fig. 5f and g). Taken together, we concluded that expression of ASXL1-MT in LT-HSCs promotes the age-related clonal expansion independent of either aging of microenvironment or ASXL-MT-expressing stromal cells in native hematopoiesis.

**ASXL1-MT induces aberrant expansion of the LT-HSC compartment through activation of Akt/mTOR pathway.** To gain insight into the mechanism underlying the growth advantage of pLT-HSCs expressing ASXL1-MT, we performed RNA-seq analysis using HSPCs from young *Vav-Cre* ASXL1-MT KI mice and littermate control mice. Geneset enrichment analysis (GSEA) suggested that Akt/mTOR pathway was activated in ASXL1-MT KI HSPCs (Fig. 4a). Consistent with the results of GSEA, intracellular flow cytometry analyses revealed that phosphorylation of Akt and S6, which indicates activation of Akt and mTORC1, respectively, were elevated in young ASXL1-MT KI HSPCs (Fig. 4b, c, Supplementary Fig. 6a, b). Next, we evaluated the age-related change of Akt/mTOR activity in *Vav-Cre* ASXL1-MT KI mice. LT-HSCs of both young and aged mice exhibited activation of Akt/mTOR signaling to the same degree in *Vav-Cre* ASXL1-MT KI mice compared with age-matched control mice (Fig. 4d and e).

As Akt/mTOR signaling positively controls cell proliferation, we inferred that aberrant proliferation of ASXL1-MT-expressing pLT-HSCs was caused by enhanced Akt/mTOR signaling. Therefore, we treated aged *Vav-Cre* ASXL1-MT KI mice and age-matched control mice with an mTOR inhibitor rapamycin for 8 weeks (Fig. 5a). Intriguingly, treatment with rapamycin reduced the frequency of pLT-HSCs in aged *Vav-Cre* ASXL1-MT KI mice (Fig. 5b and c). Similar to young mice, pLT-HSCs in G0 phase were significantly decreased in aged *Vav-Cre* ASXL1-MT KI mice, which was normalized by rapamycin treatment (Fig. 5d). In contrast, ASXL1-MT had no effect on the frequency of apoptotic LT-HSCs in aged mice (Fig. 5e). Treatment with an Akt inhibitor perifosine also normalized cell cycle status in aged *Vav-Cre* ASXL1-MT KI mice (Fig. 5f and g). These data suggest that abnormal cell cycle progression contributes to gradual expansion of the LT-HSC compartment during aging in *Vav-Cre* ASXL1-MT KI mice. Collectively, the activated Akt/mTOR pathway plays a pivotal role to confer growth advantage on LT-HSCs expressing ASXL1-MT.

**ASXL1-MT/BAP1 complex promotes AKT deubiquitination and stabilization.** Previous studies have shown that phosphorylated AKT is ubiquitinated and degraded[38–42], and wild-type

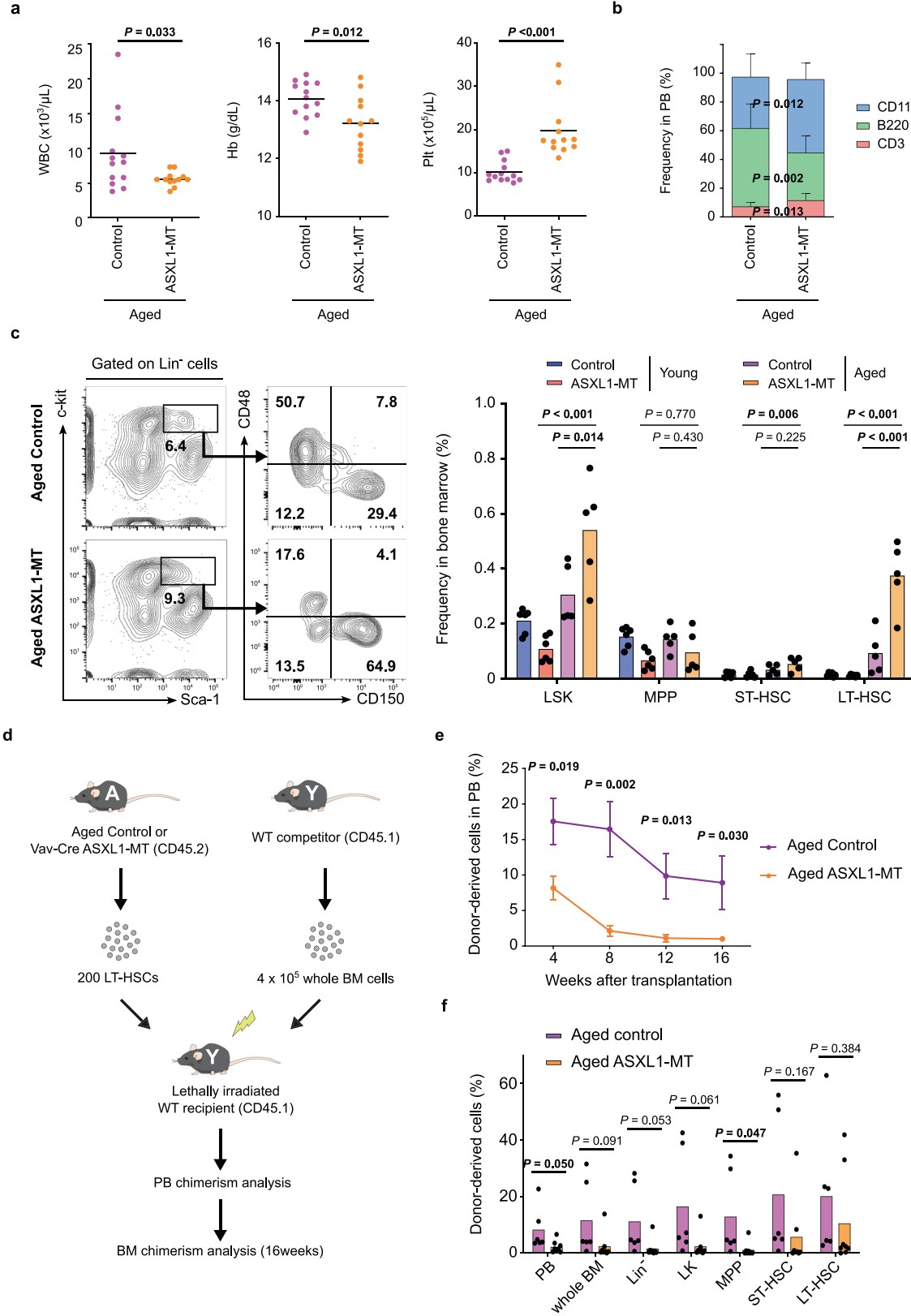

ASXL1 interacts with phosphorylated AKT[43,44]. Therefore, we first examined whether ASXL1 and its partner protein BAP1 interact with AKT. We found that wild-type/mutant ASXL1 and BAP1 bound to AKT in both cytoplasm and nucleus in 293T cells, suggesting that they form a complex (Fig. 6a, b,

and Supplementary Fig. 7a). Because mutant ASXL1 is known to stabilize BAP1 and enhance its DUB activity[23,24], we next examined whether ubiquitination of AKT is removed by the ASXL1-MT/BAP1 complex. Consistent with our previous reports[23], ASXL1-MT and BAP1 were mutually stabilized

**Fig. 2 ASXL1-MT expands the pLT-HSC compartment along with dysregulated hematopoiesis during aging. a** Enumeration of white blood cells (WBC), hemoglobin (Hb), and platelets (Plt) in peripheral blood of aged *Vav-Cre* ASXL1-MT KI mice (n = 13 (Control) and 12 (ASXL1-MT)). **b** The frequency of myeloid cells (CD11b[+]), B cells (B220[+]), and T cells (CD3[+]) in peripheral white blood cells of aged *Vav-Cre* ASXL1-MT KI mice (n = 13 (Control) and 12 (ASXL1-MT)). **c** The frequency of LSK cells, MPPs, ST-HSCs, and LT-HSCs in bone marrow cells of young and aged *Vav-Cre* ASXL1-MT KI mice (n = 6 (Young) and 5 (Aged)). Representative FACS plot (left panel) and summarized data (right panel) are shown. **d** The experimental design for competitive transplantations. 200 LT-HSCs isolated from aged *Vav-Cre* ASXL1-MT KI mice and age-matched control mice were transplanted into lethally irradiated recipient mice with $4 \times 10^5$ whole bone marrow cells. **e** Levels of donor chimerism in peripheral blood were analyzed at the indicated weeks after transplantation (n = 7 (Control) and 8 (ASXL1-MT)). **f** 6 months after transplantation, the frequency of donor-derived cells in peripheral blood (PB), whole bone marrow cells, Lin− cells, Lin−c-kit+Sca1−(LK) cells, MPPs, ST-HSCs, and LT-HSCs were analyzed (n = 6 (Control) and 8 (ASXL1-MT)). Data are mean ± s.e.m. Data are mean ± s.d. unless otherwise noted. Data are assessed by two-tailed Student's *t*-test (**a**, **b**, **e**, **f**) or one-way ANOVA with Tukey–Kramer's post-hoc test (**c**). *$P \leq 0.05$, **$P \leq 0.01$, ***$P \leq 0.001$.

(Fig. 6c). Notably, expression of BAP1 together with ASXL1-MT, but not with wild-type ASXL1, effectively deubiquitinated AKT mainly through K48-linked ubiquitin (Fig. 6c, d, and Supplementary Fig. 7b). Thus, these data suggest that AKT is a non-histone targets of the ASXL1-MT/BAP1 DUB complex.

To determine the role of endogenous Bap1 on Akt signaling, we then assessed the effect of Bap1 deletion in murine bone marrow cells transformed by combined expression of SETBP1-D868N and ASXL1-MT (cSAM cells: cells with combined expressing of mutant SETBP1 and ASXL1-MT) in which Akt is activated[45]. As described in our previous report[23], Bap1 depletion destabilized and reduced ASXL1-MT protein levels (Fig. 6e). In addition, Bap1 deletion in cSAM cells decreased expression of phosphorylated Akt (Fig. 6e). A time course experiment with IL-3 stimulation revealed attenuated and shortened phosphorylation of Akt in Bap1-depleted cSAM cells (Fig. 6f and g). These data suggest that ASXL1-MT cooperates with Bap1 to deubiquitinate and stabilize phosphorylated Akt, leading to activation of Akt/mTOR pathway.

**ASXL1-MT activates mitochondrial metabolism and alters mitochondrial dynamics.** GSEA of the RNA-seq data also suggested the aberrant activation of mitochondrial metabolism in HSPCs of *Vav-Cre* ASXL1-MT KI mice (Fig. 7a, Supplementary Fig. 8a, and Supplementary Table 1). As recent studies have shown that mitochondrial dynamics are closely linked to stem cell function[46–48], we therefore examined the possible influence of ASXL1-MT on mitochondrial activity. MitoTracker staining showed increased mitochondrial membrane potential in HSPCs of *Vav-Cre* ASXL1-MT KI mice (Fig. 7b). In addition, extracellular flux analyses revealed increased oxygen consumption rates (OCR) in c-kit+ cells of *Vav-Cre* ASXL1-MT KI mice (Fig. 7c). We then conducted ion chromatography–mass spectrometry (IC–MS)-based metabolomics to analyze intracellular metabolites using HSPCs. As shown in Supplementary Fig. 8b and c, the pool of TCA cycle-intermediates as well as levels of ATP were increased in HSPCs of *Vav-Cre* ASXL1-MT KI mice. These data suggest enhanced mitochondrial respiration in HSPCs of *Vav-Cre* ASXL1-MT KI mice. Interestingly, immuno-fluorescence staining of the mitochondrial protein Tom20 revealed that mitochondria in HSPCs of *Vav-Cre* ASXL1-MT KI mice tended to form a few large aggregates, while those in HSPCs of control mice showed a more punctuated and dispersed morphology (Fig. 7d and e). These morphological changes suggest altered mitochondrial fission and fusion in HSPCs of *Vav-Cre* ASXL1-MT KI mice[47,49,50]. Collectively, these results indicate that ASXL1-MT enhances mitochondrial activity and alters mitochondrial dynamics in HSPCs.

**ROS-mediated DNA damage causes dysfunction of HSPCs in ASXL1-MT KI mice.** Mitochondria are the major source of ROS, which can cause DNA damage[51,52]. We therefore assessed the levels of ROS in *Vav-Cre* ASXL1-MT KI mice. As expected, ASXL1-MT KI HSPCs showed increased ROS levels (Fig. 7f). Moreover, alkaline comet assay using pLT-HSCs revealed increased DNA strand breaks in young *Vav-Cre* ASXL1-MT KI mice compared to young control mice (Fig. 7g). We also observed increased nuclear foci of γ-H2AX in ASXL1-MT KI HSPCs after transplantation (Supplementary Fig. 8d). Notably, aged *Vav-Cre* ASXL1-MT KI mice harbor more DNA damage than do young *Vav-Cre* ASXL1-MT KI mice and age-matched control mice (Fig. 7g), implicating stronger age-related accumulation of DNA damage in pLT-HSCs of *Vav-Cre* ASXL1-MT KI mice.

Next, we investigated whether treatment with the antioxidant scavenger N-acetylcysteine (NAC) mitigates DNA damage and restores the impaired repopulation potential of HSPCs in *Vav-Cre* ASXL1-MT KI mice. We transplanted bone marrow cells from *Vav-Cre* ASXL1-MT KI mice or littermate control mice, treated with NAC for 8 weeks, into recipient mice (Fig. 8a). NAC treatment effectively normalized ROS and γ-H2AX levels of ASXL1-MT KI HSPCs (Fig. 8b and c), and partially restored their repopulation ability (Fig. 8d). Combined overexpression of catalase and a ROS-detoxifying enzyme, manganese superoxide dismutase (Cat/SOD2), also rescued the defective repopulation potential of HSPCs of *Vav-Cre* ASXL1-MT mice (Fig. 8f and g). Thus, reduction of ROS efficiently reverts the dysfunction of HSPCs in *Vav-Cre* ASXL1-MT KI mice.

p53 serves to maintain genomic stability through DNA damage response and regulation of ROS levels, but it can also cause dysfunction of HSCs[53–55]. To assess the influence of p53 on ASXL1-MT KI HSPCs, we crossed *Vav-Cre* ASXL1-MT KI mice with p53−/− mice. ASXL1-MT KI/p53−/− mice showed significantly higher frequency of LT-HSCs than ASXL1-MT KI mice (Supplementary Fig. 9a and b). Competitive transplantation assays revealed that loss of *p53* restored the repopulation potential of ASXL1-MT KI HSPCs (Supplementary Fig. 9c). Meanwhile, ASXL1-MT KI/p53−/− mice are prone to early onset of thymic lymphomas compared with p53−/− mice (Supplementary Fig. 9d–f). Transduction of dominant-negative form of p53 (p53DD) increased the repopulation potential of ASXL1-MT KI HSPCs associated with increased ROS levels and DNA damage (Supplementary Fig. 9g–j). These findings indicate that p53 plays a critical role in suppressing ROS levels and preventing tumor formation at the expense of HSPC functions in *Vav-Cre* ASXL1-MT KI mice, presumably by eradicating cells with critical DNA damage.

**Overactive Akt/mTOR signaling is responsible for dysregulated hematopoiesis in aged ASXL1-MT KI mice.** As mTORC1, a major downstream effector molecule of AKT signaling, regulates mitochondrial biogenesis[46,56,57], we inferred that the elevated mitochondrial activity was caused by enhanced Akt/mTOR signaling. Consistent with this hypothesis, treatment with

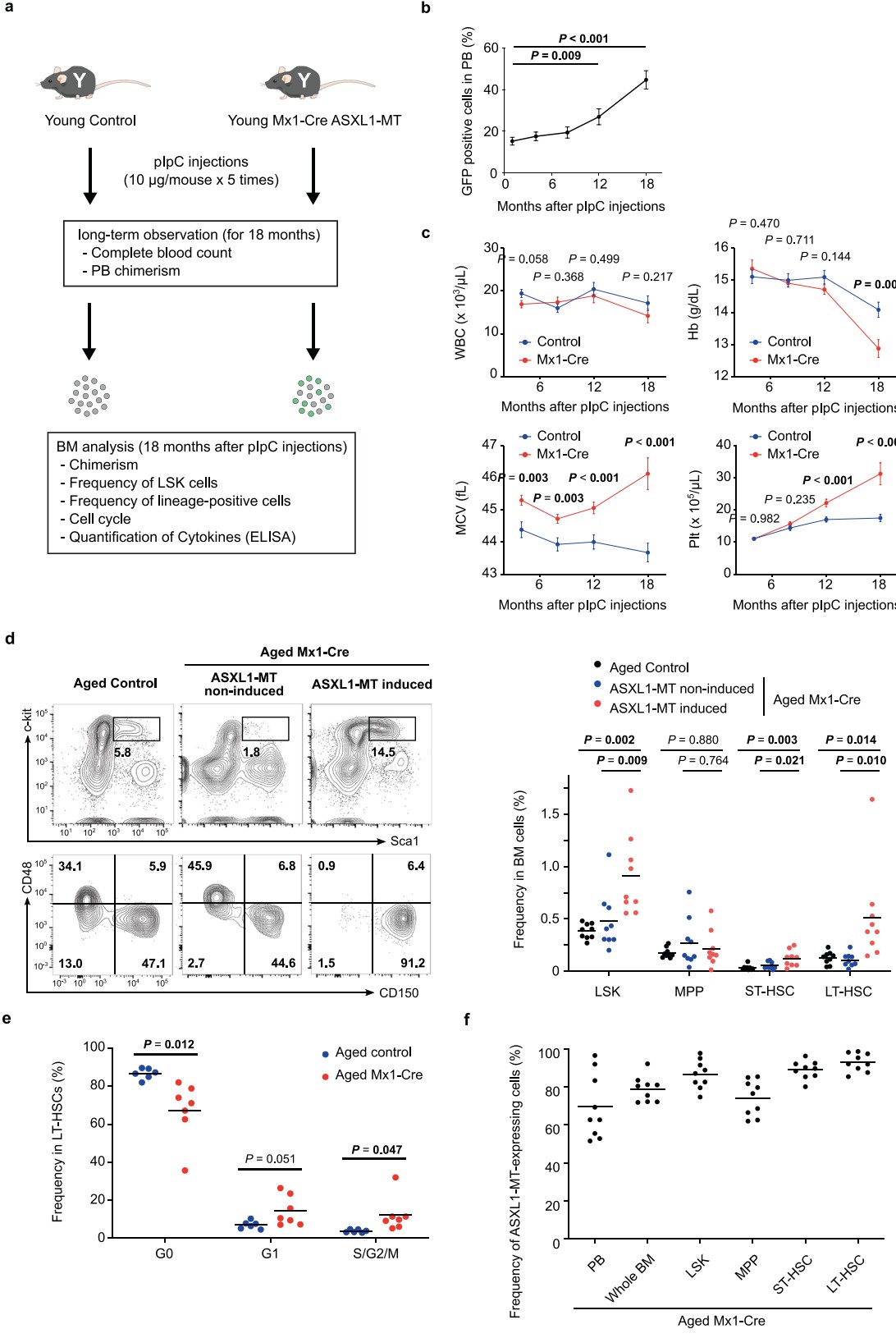

perifosine as well as rapamycin effectively decreased the mito-chondrial membrane potential and intracellular ROS levels in HSPCs of young *Vav-Cre* ASXL1-MT KI mice (Supplementary Fig. 10a–f). We next assessed whether the enhanced Akt/mTOR signaling reduced engraftment of ASXL1-MT KI HSPCs. Bone

marrow cells from young *Vav-Cre* ASXL1-MT KI mice or lit-termate control mice treated with rapamycin were transplanted into recipient mice (Supplementary Fig. 10d). Rapamycin treat-ment effectively improved engraftment of ASXL1-MT-expressing bone marrow cells (Supplementary Fig. 10g). These data indicate

**Fig. 3 ASXL1-MT confers a fitness advantage on LT-HSCs with aging. a** The experimental design for partial induction of ASXL1-MT in vivo using *Mx1-Cre* ASXL1-MT KI mice. Young *Mx1-Cre* ASXL1-MT KI mice were partially induced to express ASXL1-MT by pIpC injections (10 μg/mouse × 5 times) at 12 weeks after birth. 18 months after pIpC injections, bone marrow cells from *Mx1-Cre* ASXL1-MT KI mice were analyzed. **b** The frequency of ASXL1-MT expressing cells in peripheral blood at the indicated months after pIpC injections ($n = 20$). **c** Enumeration of white blood cells (WBC), hemoglobin (Hb), mean corpuscular volume (MCV), and platelets (Plt) in peripheral blood of *Mx1-Cre* ASXL1-MT KI mice at the indicated months after pIpC injections ($n = 20$). **d** The frequency of LSK cells, MPPs, ST-HSCs and LT-HSCs in ASXL1-MT induced cells, ASXL1-MT non-induced cells, or whole bone marrow cells of age-matched control mice ($n = 9$). Representative FACS plot (left panel) and summarized data (right panel) are shown. **e** Cell cycle analysis with Ki-67/DAPI staining of LT-HSCs of aged *Mx1-Cre* ASXL1-MT KI mice and age-matched control mice ($n = 6$). **f** The frequency of ASXL1-MT-expressing cells in peripheral blood, whole bone marrow, LSK, MPP, ST-HSC, and LT-HSC fractions of aged mice (18 months after pIpC injections) ($n = 9$). Data are mean ± s.e.m. Data are assessed by two-tailed Student's *t*-test (**b**, **c**, **e**) or one-way ANOVA with Tukey–Kramer's post-hoc test (**d**). *$P \leq 0.05$, **$P \leq 0.01$, ***$P \leq 0.001$.

that enhanced Akt/mTOR-signaling induced by ASXL1-MT provokes mitochondrial activation, ROS overproduction, and dysfunction of HSPCs. It should be noted, however, that neither antioxidant nor rapamycin inhibits the myeloid-skewed hematopoiesis in young *Vav-Cre* ASXL1-MT KI mice (Fig. 8e and Supplementary Fig. 10h). Therefore, it is likely that ASXL1-MT promotes myeloid skewing through Akt/mTOR-independent functions.

To examine the relationship between the enhanced Akt/mTOR signaling and dysregulated hematopoiesis in aged ASXL1-MT KI mice, we treated aged *Vav-Cre* ASXL1-MT KI mice and age-matched control mice with rapamycin for 8 weeks (Fig. 9a). As was the case with young mice, mitochondrial membrane potential and intracellular ROS levels were increased in LT-HSCs of aged *Vav-Cre* ASXL1-MT KI mice compared with age-matched control mice (Fig. 9b and c). ASXL1-MT-induced elevation of mitochondrial membrane potential and ROS levels was further validated in LT-HSCs of aged *Mx1-Cre* ASXL1-MT KI mice (Supplementary Fig. 11a–c). Rapamycin treatment tended to reduce the mitochondrial membrane potential and normalized ROS levels in aged LT-HSCs expressing ASXL1-MT (Fig. 9b and c). Treatment with perifosine also abrogated mitochondrial activation in these cells (Supplementary Fig. 11d and e). Importantly, treatment with rapamycin recovered the number of differentiated white blood cells as well as bone marrow cellularity in aged *Vav-Cre* ASXL1-MT KI mice (Fig. 7d and e). Furthermore, mitochondrial activation, increased DNA damage, and differentiation defects in aged ASXL1-MT KI HSPCs were ameliorated by rapamycin treatment (Fig. 9f–j). Taken together, these data suggest that overactive Akt/mTOR pathway causes dysregulated hematopoiesis, whereas it confers a growth advantage on pLT-HSCs in aged ASXL1-MT KI mice.

**ASXL1-MT enforces age-associated patterns of gene expression in LT-HSCs.** To further characterize the impact of ASXL1-MT on physiological aging of LT-HSCs, we performed RNA-seq analysis using LT-HSCs from aged *Vav-Cre* ASXL1-MT KI mice and age-matched control mice (Fig. 10a and b). As was the case with young mice, GSEA suggests activation of Akt/mTOR pathway in aged *Vav-Cre* ASXL1-MT KI mice, in concordance with flow cytometry analyses (Figs. 4d, e, and 10c). In aged ASXL1-MT-expressing LT-HSCs, up-regulated genes significantly overlapped with genesets related to Akt/mTOR signaling, platelet signature, and senescence; conversely, down-regulated genes significantly overlapped with genesets related to myeloid differentiation, erythroid differentiation, and HSC signature (Fig. 10d). Since ASXL1-MT appears to accelerate aging of LT-HSCs (e.g. anemia, myeloid-skewed differentiation, and hypocellular bone marrow), we compared alteration of gene expression profiles associated with physiological aging of LT-HSCs (GSE48893) with this RNA-seq analysis data. This analysis revealed that ASXL1-MT promotes the expression of age-associated patterns, suggesting that ASXL1-MT accelerates aging of LT-HSCs (Fig. 10e and f).

**AKT/mTOR pathway seems to be activated in MDS patients harboring *ASXL1* mutations compared with those harboring *DNMT3A* or *TET2* mutations.** Finally, we examined whether the development of MDS, a clonal HSC disease frequently arising from CH, harboring *ASXL1* mutations is associated with enhanced Akt/mTOR signaling. To this end, we analyzed a public transcriptome data of CD34-positive bone marrow cells from MDS patients with *DNMT3A*, *TET2*, or *ASXL1* mutations, and healthy control subjects (Gene Expression Omnibus accession number: GSE58831). GSEA suggested that AKT pathway is up-regulated in MDS patients harboring *ASXL1* mutations, but not those harboring *DNMT3A* or *TET2* mutations, compared to healthy control subjects (Supplementary Table 2). We also found downregulation of negatively regulated genes by mTOR signaling in MDS patients with *ASXL1* mutations (Supplementary Table 2). These data suggest that AKT/mTOR pathway is activated in MDS patients harboring *ASXL1* mutations.

## Discussion

Somatic mutations in epigenetic regulators, *ASXL1*, *TET2*, and *DNMT3A*, are recurrently detected in CH[19–21]. *TET2* or *DNMT3A* deficiency in mice enhances self-renewal of LT-HSCs[30–33], suggesting that mutations in *TET2* and *DNMT3A* induce clonal expansion of hematopoietic cells, leading to CH in humans. In contrast, *ASXL1* mutations in mice reduce the number and function of HSCs[25,34,35]. In the present study, we confirmed that ASXL1-MT causes a competitive disadvantage of LT-HSCs after transplantation. Nevertheless, LT-HSCs expressing ASXL1-MT acquire a growth advantage and eventually occupy the HSC compartment during aging in genetic mosaic mouse model. This observation is clearly distinct from that of *DNMT3A* or *TET2* mutations, which exhibits an increase in the long-term regenerative potential of HSCs. Our results provide an insight into the pathogenesis of CH, in that CH can be induced by clonal expansion of not only "true" HSCs but also by long-lived pLT-HSCs lacking repopulation potential[1,9]. The enhanced Akt/mTOR pathway is expected to provoke replication stress along with increased ROS-mediated DNA damage. Indeed, we observed increased DNA strand breaks in aged ASXL1-MT KI pLT-HSCs, indicating that ASXL1-MT expands the pLT-HSC compartment with accumulation of DNA damage. This could be one mechanistic explanation for why individuals with CH are at increased risk for subsequent hematological malignancies, associated with secondary mutations.

The present results suggest that the activated Akt/mTOR pathway, which promotes cell cycle progression, is responsible for aberrant expansion of the LT-HSC compartment in aged ASXL1-MT KI mice. At the same time, we also found that overactive Akt/mTOR signaling induced by ASXL1-MT provokes dysfunction of HSCs. It has been shown that the enhanced Akt/mTOR activity by depletion of *Pten* results in transient expansion and subsequent depletion of HSCs[58,59]. It appears that ASXL1-MT has a much milder impact on Akt/mTOR pathway than disruption of a

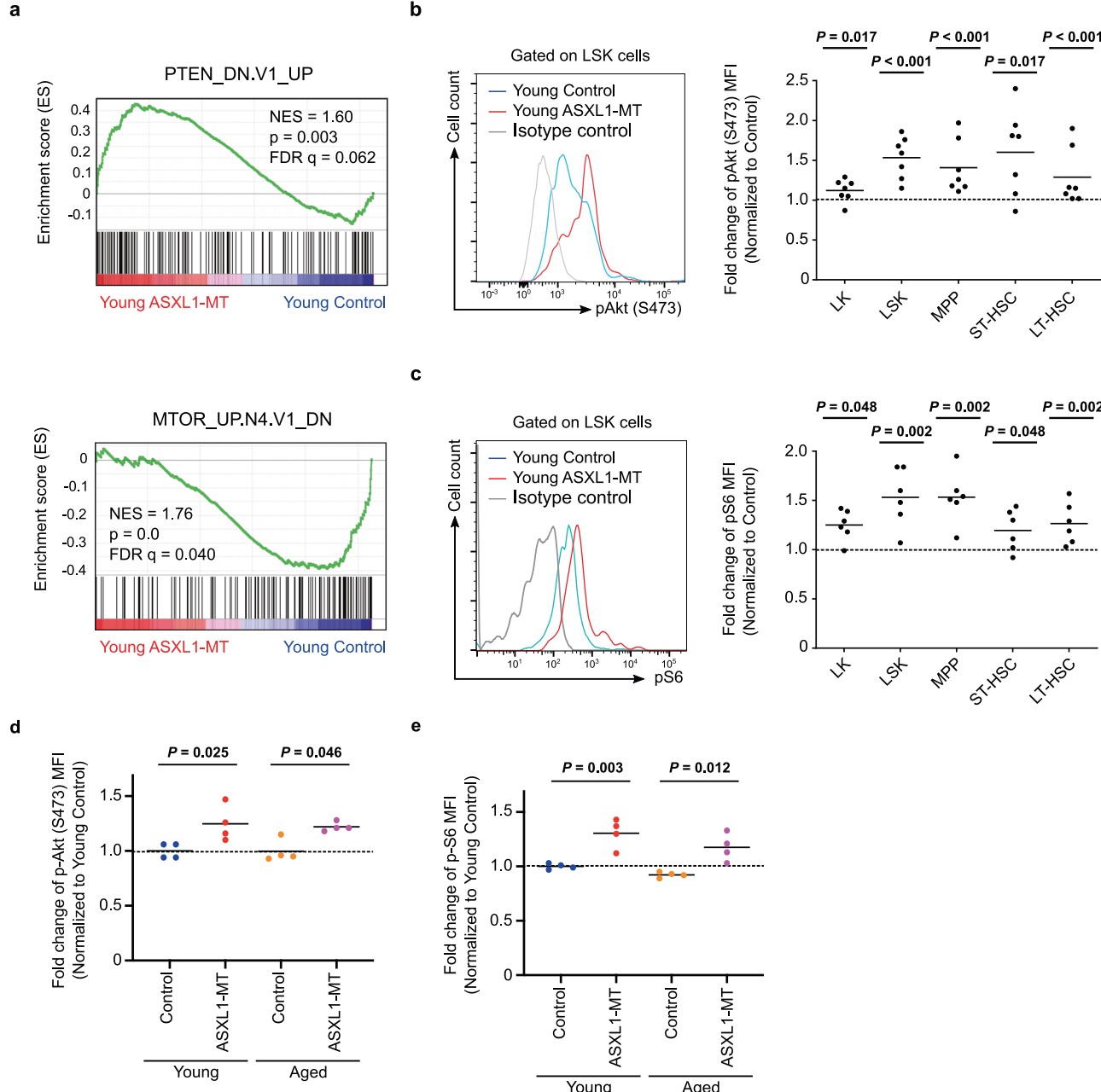

**Fig. 4 ASXL1-MT activates Akt/mTOR pathway in both young and aged mice. a** Gene set enrichment analysis of LSK cells from young *Vav-Cre* ASXL1-MT KI mice and littermate control mice using genesets up-regulated by PTEN deficiency (PTEN_DN.V1_UP) and negatively regulated by mTOR signaling (MTOR_UP.N4.V1_DN)(*n* = 3). **b, c** Levels of phosphorylated Akt (S473) (**b**) and phorphorylated S6 (**c**) in LK cells, LSK cells, MPPs, ST-HSCs and LT-HSCs from young *Vav-Cre* ASXL1-MT KI mice (*n* = 7). Representative FACS plot (left panel) and summarized data (right panel) are shown. **d, e** Levels of phosphorylated Akt (S473) (**d**) and phosphorylated S6 (**e**) in LT-HSCs of young and aged *Vav-Cre* ASXL1-MT KI mice (*n* = 4). Statistical significances are assessed by two-tailed Mann–Whitney's *u*-test (**b, c**) or one-way ANOVA with Tukey–Kramer's post-hoc test (**d, e**). \**P* ≤ 0.05, \*\**P* ≤ 0.01, \*\*\**P* ≤ 0.001.

core repressive component comprising Akt/mTOR pathway. The modestly activated AKT/mTOR signaling induced by ASXL1-MT presumably has negative effects on HSC function, whereas it causes abnormal expansion of pLT-HSCs in steady-state hematopoiesis over time. These molecular changes should underlie the expansion of pLT-HSCs along with impaired repopulation potential in ASXL1-MT KI mice. Related to these findings, we observed enhancement of age-associated phenotypes in hematopoiesis including anemia, myeloid-biased differentiation, hypocellular bone marrow, and expansion of surface marker-defined LT-HSCs in aged ASXL1-MT KI mice. These phenotypes were partly rescued by inhibition of Akt/mTOR pathway (Figs. 5c and

9d), suggesting that ASXL1-MT-induced activation of Akt/mTOR pathway is involved in enhanced aging of the hematopoietic system. In addition, RNA-seq analyses revealed that ASXL1-MT promotes the gene expression of age-associated patterns. These results indicate that ASXL1-MT could promote HSC aging via activation of Akt/mTOR pathway.

One of the most intriguing findings of this work is the identification of AKT as a non-histone target of BAP1. BAP1, which is stabilized and given an enhanced DUB activity by ASXL1-MT, is a known DUB for histone H2A, and was also shown to deubiquitinate several non-histone proteins[60–64]. It was previously demonstrated that multiple E3 ligases induce ubiquitination and

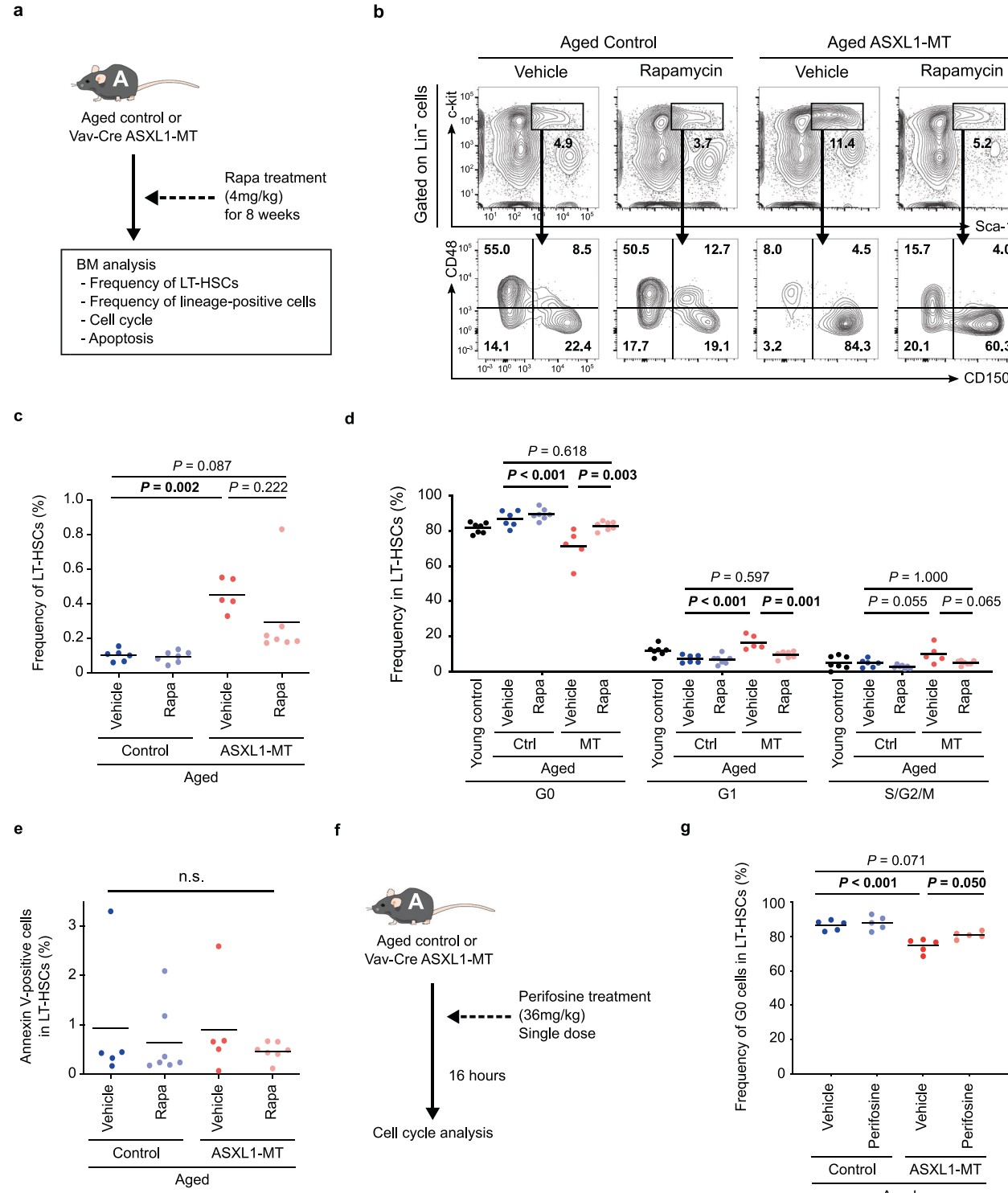

**Fig. 5 Inhibition of Akt/mTOR pathway ameliorates ASXL1-MT-induced aberrant proliferation of pLT-HSCs. a** The experimental design for treatment with an mTOR inhibitor rapamycin in aged mice. Aged *Vav-Cre* ASXL1-MT KI mice and age-matched control mice were treated with rapamycin intraperitoneally (4 mg/kg/day) every other day for 8 weeks. At the end of administrations, end-point analyses of bone marrow cells were conducted. **b**, **c** The frequency of LT-HSCs in bone marrow MNCs ($n = 6$ (Control-Vehicle), 7 (Control-Rapamycin), 5 (ASXL1-MT-Vehicle), and 7 (ASXL1-MT-Rapamycin)). Representative FACS plot (**b**) and summarized data (**c**) are shown. **d** Cell cycle analysis with Ki-67/DAPI staining of LT-HSCs ($n = 7$ (Young control), 6 (Control-Vehicle), 7 (Control-Rapamycin), 5 (ASXL1-MT-Vehicle), and 7 (ASXL1-MT-Rapamycin)). **e** Apoptosis analyses of LT-HSCs ($n = 5$ (Control-Vehicle), 7 (Control-Rapamycin), 5 (ASXL1-MT-Vehicle), and 7 (ASXL1-MT-Rapamycin)). **f** The experimental design for treatment with an Akt inhibitor perifosine in aged mice. Aged *Vav-Cre* ASXL1-MT KI mice and age-matched control mice were treated with a single oral dose (36 mg/kg) of perifosine ($n = 5$). **g** 16 h after administration, cell cycle analyses with Ki-67/DAPI staining of LT-HSCs were performed. Statistical significances are assessed by one-way ANOVA with Tukey–Kramer's post-hoc test. *$P \leq 0.05$, **$P \leq 0.01$, ***$P \leq 0.001$.

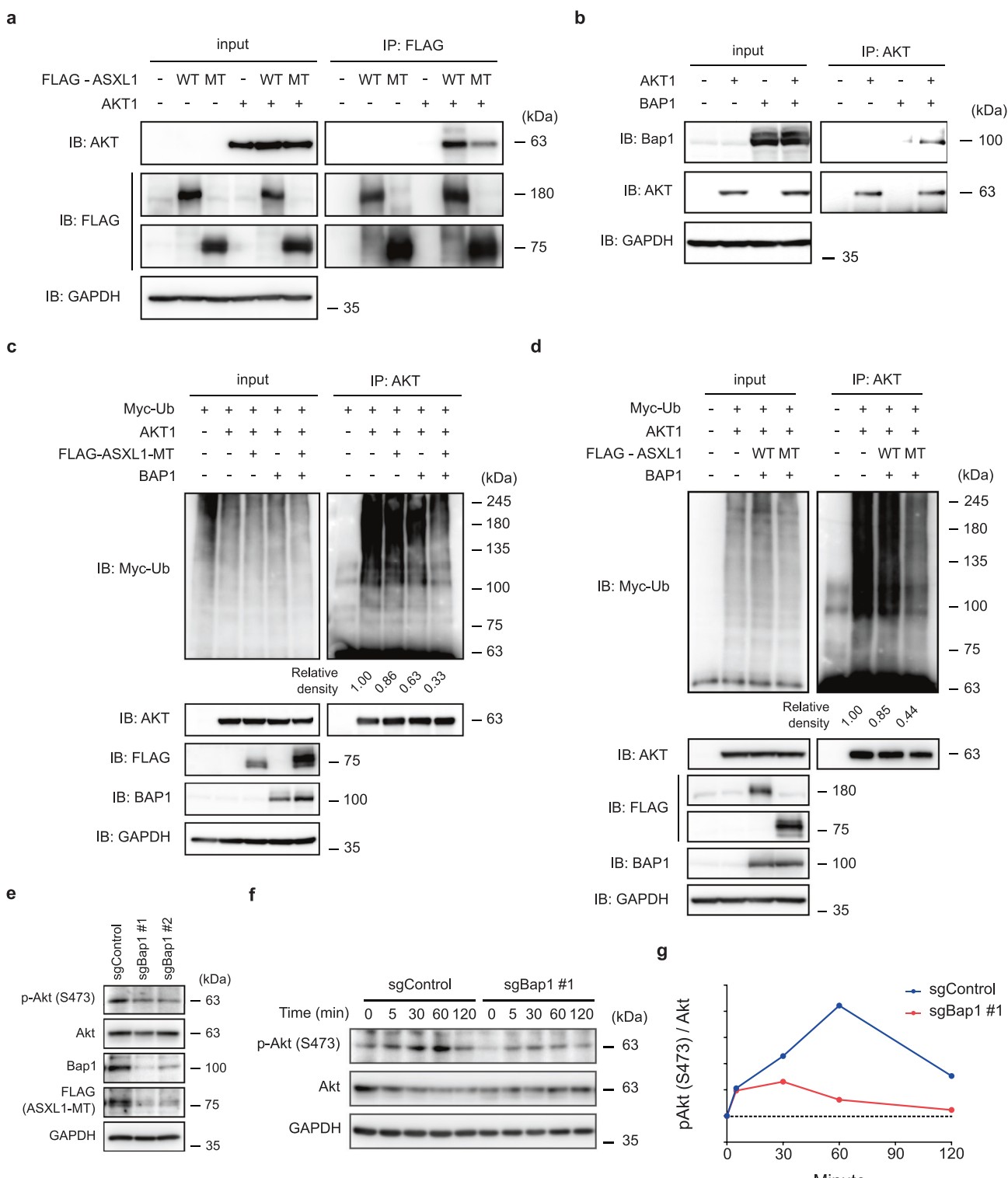

degradation of phosphorylated AKT, thereby quenching activated AKT signaling[38,40–42]. Conversely, ASXL1-MT/BAP1 complex deubiquitinates and stabilizes AKT, which leads to weak but possibly continuous activation of Akt/mTOR pathway. Thus, these findings revealed a function of ASXL1-MT as a signaling modulator beyond its known roles as an epigenetic regulator. On the other hand, one recent study has shown that *Asxl1* loss activates Akt/mTOR pathway due to an epigenetic dysregulation at the *Pten* locus[65]. Thus, we cannot completely exclude the possibility that

ASXL1-MT activates Akt/mTOR pathway through an epigenetic-dependent manner as well.

Among individuals with CH, the increased risk for developing hematological malignancies and cardiovascular diseases is associated with higher variant allele frequency[21,66]. Thus, it is possible to avoid the onset of these diseases by inhibiting the expansion of mutated CH clone. In the present study, we demonstrate that the aberrant cell cycle progression and the increased frequency of pLT-HSCs are suppressed by treatment with rapamycin in aged

**Fig. 6 ASXL1-MT/BAP1 complex deubiquitinates and stabilizes Akt. a** 293T cells were transfected with FLAG-ASXL1-WT, FLAG-ASXL1-MT, and AKT1 expressing vectors. Total cell lysates were subjected to immunoprecipitation using anti-FLAG antibody followed by immunoblotting. **b** 293T cells were transfected with AKT1 and BAP1 expressing vectors. Total cell lysates were subjected to immunoprecipitation using anti-AKT antibody followed by immunoblotting. **c** 293T cells were transfected with AKT1, FLAG-ASXL1-MT, BAP1, and Myc-Ubiquitin expressing vectors. Total cell lysates were subjected to immunoprecipitation using anti-AKT antibody followed by immunoblotting. Relative levels of Myc-Ubiquitin were quantified by densitometry and normalized to total Myc-Ubiquitin levels. **d** 293T cells were transfected with AKT1, FLAG-ASXL1-WT, FLAG-ASXL1-MT, BAP1, and Myc-Ubiquitin expressing vectors. Total cell lysates were subjected to immunoprecipitation using anti-AKT antibody followed by immunoblotting. Relative levels of Myc-Ubiquitin were quantified by densitometry and normalized to total Myc-Ubiquitin levels. **e–g** Murine bone marrow cells transformed by combined expression of SETBP1-D868N and ASXL1-MT (cSAM cells) were transduced with Cas9 and sgRNA targeting Bap1 (**e**). Bap1-depleted cSAM cells were then starved of IL-3 for 3 h followed by stimulation with IL-3 (1 ng/mL) for the indicated times (**f**). Relative levels of phosphorylated Akt were quantified by densitometry and normalized to total Akt levels (**g**). Similar result was obtained from Bap1-depleted cSAM cells using another sgRNA-targeting BAP1 (sgBAP1 #2). A representative experiment from at least $n = 2$ independent experiments is shown.

ASXL1-MT KI mice. Hence, a pharmacological inhibition of the Akt/mTOR pathway to individuals with CH harboring ASXL1 mutations is expected to reduce a clonal advantage of the CH clone.

Besides CH, recent reports have revealed that normal tissues also accumulate somatic mutations with aging, which may confer a selective advantage[67]. However, mutational landscapes are clearly distinct; while mutations in epigenetic factors and splicing factors are frequently detected in CH, most mutations detected in oncogenes are signal transduction molecules in other tissues. At present, the reason for this discrepancy remains totally elusive. Clarification of the mechanisms by which CH progress to hematological malignancies may provide us a clue to understand the difference.

In conclusion, we show that ASXL1-MT collaborates with BAP1 to activate Akt/mTOR pathway in an epigenetics-independent manner. The activated Akt/mTOR pathway causes aberrant expansion of pLT-HSCs to occupy the HSC compartment during aging. Overactive Akt/mTOR signaling also causes mitochondrial activation, overproduction of ROS, increased DNA damage, and subsequent dysfunction of HSCs. The expansion of pLT-HSCs along with increased DNA damage caused by ASXL1 mutations can result in the development of CH, leading to hematopoietic malignancy with secondary mutations. A pharmacological inhibition of Akt/mTOR pathway may pave the way for a preventive intervention to individuals with CH harboring ASXL1 mutations.

## Methods
**Mice**. Wild-type C57BL/6J mice were bred in-house. $p53^{-/-}$ mice were provided by K. Matsuda (Tokyo University). Conditional ASXL1-MT KI mice have been generated in our laboratory and crossed to *Vav-Cre* or *Mx1-Cre* transgenic mice. Experiments were performed with 6–12-week-old for young mice or 20–24-month-old for aged mice. All mice were housed with a 12 h dark/light cycle at a temperature between 20–25 °C and a humidity between 40% and 60%. The experiments were approved by the Committee on the Ethics of Animal Experiments and all these mice were maintained according to the guidelines of the Institute of Laboratory Animal Science (PA13–19 and PA16–31).

**N-acetyl-ʟ-cysteine treatment**. To examine the effect of ROS on repopulation ability of HSPCs, ASXL1-MT KI mice and littermate control mice were treated with drinking water containing 1 mg/mL N-acetyl-ʟ-cysteine (NAC) (SIGMA, A7250) for 8 weeks before transplantation. Recipient mice were continuously treated with NAC after transplantation.

**Rapamycin treatment**. Rapamycin (LC Laboratories, R-5000) was dissolved in DMSO (20 mg/mL) and further diluted with 10% PEG-300 (SIGMA, 202371)/5% Tween 80 (SIGMA, P4780)/PBS. Mice were treated with 4 mg/kg rapamycin intraperitoneally as described in figure legends.

**Perifosine treatment**. Perifosine (Chemscene LLC, CS-0209) was dissolved in sterile water, and 36 mg/kg was administered orally 16 h before analyses.

**polyinosine–polycytidine (pIpC) treatment**. *Mx1-Cre* ASXL1-MT KI mice were intraperitoneally injected with 250 μg pIpC (SIGMA, P1530) three times every

other day for competitive transplantations, or 10 μg pIpC 5 times every other day for partial induction of ASXL1-MT at 12 weeks after birth.

**Plasmids**. The SF91-IRES-GFP and SF91-Cat/SOD2-IRES-GFP vectors were kindly provided by Michael D. Milsom. SF91-IRES-Venus and SF91-Cat/SOD2-IRES-Venus vectors were generated by replacing GFP with Venus sequence at the PmlI/BsrGI sites. The 3xFLAG-tagged ASXL1-WT and ASXL1-MT (1900–1922del; E635RfsX15) were cloned into pMYs-IRES-GFP vector as previously described[23]. The HA-tagged BAP1 was cloned into pMYs-IRES-NGFR vector as previously described[23]. T7-p53DD-pcDNA3 (Addgene #25989) was obtained from Addgene, and we cloned it to pMYs-IRES-NGFR vector as previously described[68]. The HA-tagged AKT1 was cloned into pMXs-IRES-GFP vector at the BamHI/EcoRI sites.

**Cell culture**. 293T cells (CRL-11268, ATCC, Manassas, VA, USA) were cultured in Dulbecco's modified Eagle's medium (DMEM) supplemented with 10% fetal bovine serum (FBS) and 1% penicillin/streptomycin. cSAM cells were generated by transducing SETBP1-D868N and ASXL1-MT into murine bone marrow progenitors and were cultured in RPMI1640 medium supplemented with 10% FBS, IL-3 (1 ng/mL), and 1% penicillin/streptomycin[45].

**Retroviral bone marrow transplantation**. Plat-E packaging cells were transfected with retroviral constructs using calcium phosphate co-precipitation method. After transfection, retroviral supernatant was collected and loaded onto RetroNectin-coated plates. Freshly isolated Lin⁻ cells were cultured in Iscove's modified Dulbecco's media (IMDM) supplemented with 20% FBS, 2 mM ʟ-glutamine, SCF (50 ng/mL), Flt3-ligand (50 ng/mL), and TPO (50 ng/mL) overnight, and were subsequently transduced with the viruses for 48 h.

**CRISPR/Cas9-mediated gene knockout**. To obtain short guide RNA (sgRNA) constructs targeting for Bap1, annealed oligos were cloned into lentiGuide-puro vector (Addgene, #52963) as previously described[23]. 293T cells were transfected with Cas9-expressing vector (lentiCas9-Blast; Addgene, #52962) or sgRNA-expressing vector together with lentiviral packaging vectors (pMD2.G; Addgene, #12259 and psPAX2; Addgene, #12260) using polyethylenimine (PEI), and then lentiviral supernatant was collected. cSAM cells were transduced with the viruses for 24 h and were subsequently subjected to drug selection with 10 μg/mL blasticidin for stable expression of Cas9 and 1.5 μg/mL puromycin for stable expression of sgRNA.

**Flow cytometry**. Bone marrow cells were obtained by flushing long bones (femurs and tibias) in phosphate-buffered saline (PBS) containing 2% heat-inactivated FBS (FACS buffer). Cell suspensions were lysed with erythrocyte lysis buffer (150 mM NH₄Cl, 10 mM KHCO₃, 100 μM EDTA-Na₂), filtered through a 40 μm filter, and were incubated with a cocktail of biotinylated monoclonal antibodies to lineage markers (CD5, B220, CD11b, Gr-1, and Ter119) and anti-biotin microbeads (Miltenyi Biotec). Lin⁻ cells were purified using MACS separation LS columns (Miltenyi Biotec). Cells were then stained with CD150-PE (BioLegend, 115904), c-kit-PE-Cy7 (BioLegend, 105814), Sca1-APC (BioLegend, 108112), CD48-Brilliant Violet 421 (BioLegend, 103427), and Streptavidin-Brilliant Violet 605 (BioLegend, 405229). Expression levels of CD41 were analyzed using CD41-APC (BioLegend, 133914) antibody. Peripheral blood was analyzed using CD11b-PE (eBiosciense, 12-0112-85), CD45R/B220-PE/Cy7 (BioLegend, 103224), CD3-APC (100236), and CD45.2-APC/Cy7 (BioLegend, 109824) antibodies. For ROS, mitochondrial membrane potential and intracellular staining analyses, Sca1-Brilliant Violet 785 (BioLegend, 108139) antibody was used instead of Sca1-APC antibody. For apoptosis, cell cycle and CD41 analyses, CD48-APC-Cy7 (BioLegend, 103431) and Sca1-Brilliant Violet 785 antibodies were used instead of CD48-Brilliant Violet 421 and Sca1-APC antibodies. Propidium iodide (PI) or 4′,6-diamidino-2-phenylindole (DAPI) was used to exclude dead cells. All data were collected using a FACSVerse

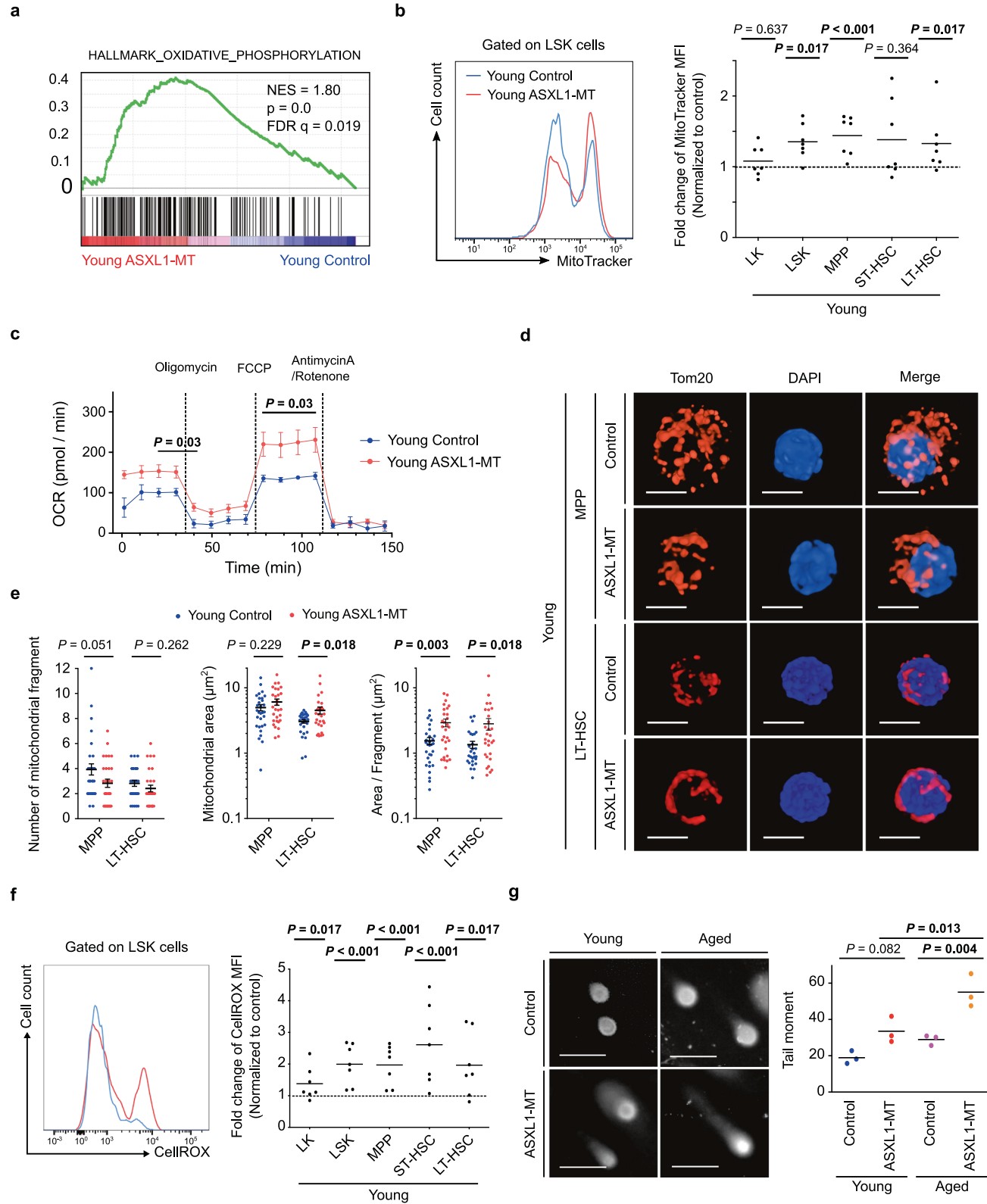

with BD FACSuite software or FACSAria with FACSDiva software and were analyzed with FlowJo software.

**Intracellular staining**. Bone marrow cells stained for surface markers were fixed in 4% paraformaldehyde for 10 min at room temperature, permeabilized with 0.2% Triton X-100 for 15 min at room temperature and blocked with 5% goat serum for 15 min at room temperature. Cells were then incubated with anti-p-H2A.X (Cell Signaling Technology, #9718, 1:200), anti-Akt (Pan) (Cell Signaling Technology, #4691, 1:400), anti-Phospho-Akt (Ser473) (Cell signaling Technology, #4060, 1:400), anti-pS6 (Ser235/236) (Cell Signaling Technology, #4858, 1:100), or isotype control (abcam, ab172730) antibody for 1 h at room temperature, respectively. Cells were then washed and incubated with Alexa Fluor 647 goat anti-rabbit IgG (Invitrogen, A21245, 1:2000) for 1 h at room temperature.

**Fig. 7 ASXL1-MT induces mitochondrial activation, overproduction of ROS, and increased DNA damage. a** Gene set enrichment analysis of TCA cycle and respiratory electron transport gene sets in LSK cells of young control and *Vav-Cre* ASXL1-MT KI mice ($n = 3$). **b** Mitochondrial membrane potential in LK cells, LSK cells, MPPs, ST-HSCs and LT-HSCs of young control and *Vav-Cre* ASXL1-MT KI mice ($n = 7$). Representative histogram in LSK cells is shown (left panel). Summarized results are expressed as fold change relative to their respective counterparts in littermate controls (right panel). **c** Oxygen consumption rates (OCR) in pooled $3 \times 10^5$ Lin−c-kit+ cells under basal conditions and in response to oligomycin, fluoro-carbonyl cyanide phenylhydrazone (FCCP) and antimycinA/rotenone. Cells were obtained from three animals per genotype. Data are mean ± s.e.m. **d** Representative immunofluorescence for Tom20 in MPPs and LT-HSCs (scale bars, 5 µm). **e** Quantification of mitochondrial fragment (left panel), area (middle panel), and area normalized to one fragment (right panel) (30 cells/group). Cells were obtained from 2 animals per genotype. Data are mean ± s.e.m. **f** Intracellular ROS levels in LK cells, LSK cells, MPPs, ST-HSCs, and LT-HSCs of young control and *Vav-Cre* ASXL1-MT KI mice ($n = 7$). Representative histogram in LSK cells is shown (left panel). Summarized results are expressed as fold change relative to their respective counterparts in littermate controls (right panel). **g** Comet analyses using LT-HSCs from young and aged *Vav-Cre* ASXL1-MT KI mice ($n = 3$). Representative images (scale bars, 20 µm) (left panel) and olive tall moment (right panel) are shown. Statistical significances are assessed by two-tailed Mann–Whitney's *u*-test (**b**, **f**), two-tailed Student's *t*-test (**c**,s **e**), or one-way ANOVA with Tukey–Kramer's post-hoc test (**g**). *$P \leq 0.05$, **$P \leq 0.01$.

**Cell cycle analysis**. Bone marrow cells stained for surface markers were fixed in 4% paraformaldehyde for 10 min at room temperature, permeabilized with 0.2% Triton X-100 for 15 min at room temperature and blocked with 5% goat serum for 15 min at room temperature. Cells were then stained with the Ki-67-eFluor660 (eBiosciense, 50-5698-82) and DAPI for 30 min at 4 °C.

**Apoptosis analysis**. Bone marrow cells were labeled with surface marker antibodies, as described previously, and were then stained with Annexin V-APC (BioLegend, 640920) antibody in binding buffer (10 mM Hepes (pH 7.4), 140 mM NaCl, 2.5 mM CaCl₂) for 30 min at room temperature.

**HSPC transplantation assay**. Lethally irradiated (900 cGy total, divided into two doses of 450 cGy given over 4 h) C57BL/6 recipient mice (CD45.1) were transplanted with either 200 LT-HSCs or 3000 MPPs (CD45.2) in competition with $4.0 \times 10^5$ C57BL/6 bone marrow mononuclear cells (MNCs) (CD45.1).

**Serial transplantation assay**. $1.0 \times 10^6$ bone marrow MNCs harvested from ASXL1-MT KI mouse were mixed with $1.0 \times 10^6$ bone marrow MNCs harvested from littermate control mouse and injected intravenously into lethally irradiated C57BL/6 recipient mice. 6 months after transplantation, $6 \times 10^6$ million bone marrow MNCs were isolated from three recipient mice and serially transplanted into lethally irradiated recipients. Donor-derived cells were evaluated by GFP positivity.

**Homing assay**. $5 \times 10^4$ LSK cells harvested from *Vav-Cre* ASXL1-MT KI mice were stained with 1 µM CellTrace Far Red (Thermo Fisher Scientific, C34564) for 20 min at room temperature. Cells were then incubated with IMDM, washed, and injected into lethally irradiated mice. 16 h after transplantation, bone marrow cells isolated from femurs and tibias of recipient mice were analyzed to count CellTrace-positive cells.

**Quantification of cytokines**. The levels of TNF-α and IFN-γ in blood serum and bone marrow fluids were measured with TNF-alpha Quantikine ELISA Kit (R&D Systems, MTA00B) and IFN-gamma Quantikine ELISA Kit (R&D Systems, MIF00), respectively, according to the manufacturer's protocol.

**Analysis of ROS levels**. For intracellular ROS analysis, bone marrow cells were incubated for 30 min at 37 °C with 5 µM CellROX Deep Red (Thermo Fisher Scientific). Cells were then washed and analyzed by flow cytometry.

**Quantification of mitochondrial membrane potential**. Bone marrow cells were incubated with 100 µM MitoTracker Red (Thermo Fisher Scientific) for 15 min at 37 °C. Cells were then washed and analyzed by flow cytometry.

**Measurement of OCR**. Sort-purified $3 \times 10^5$ Lin−c-kit+ cells were immobilized to XF24 cell culture microplates pretreated with CELL-TAK (CORNING). Mitochondrial respiration was measured with a XF24 extracellular flux analyzer and XF cell Mito Stress Test Kit (Agilent). OCR was measured under basal conditions, followed by sequential injection of the ATP synthase inhibitor oligomycin (0.5 µM), the uncoupler FCCP (trifluorocarbonylcyanide phenylhydrazone; 1.0 µM), the mitochondrial complex III inhibitor antimycin A (0.5 µM) and mitochondrial complex I inhibitor rotenone (0.5 µM).

**Immunofluorescence**. MPPs or LT-HSCs were sorted into the slide coated with poly-L-lysine (SIGMA). Cells were placed on ice for 30 min to permit cells to settle onto the slide and fixed with 4% paraformaldehyde for 20 min at room temperature. Cells were then permeabilized with 0.2% TritonX-100 for 15 min at room

temperature and blocked with 5% goat serum for 1 h at room temperature. Cells were incubated with the anti-p-H2A.X (Cell Signaling Technology, #9718, 1:200) or anti-Tom20 (Santa Cruz Biotechnology, sc-11415, 1:500) antibody overnight at 4 °C, washed and incubated in the Alexa Fluor 568 goat anti-rabbit IgG (Invitrogen, A11011, 1:1000) for 1 h at room temperature. Cell nuclei were stained with DAPI and mounted using fluorescence mounting medium (DAKO). Images were acquired with a Confocal Microscope A1R or super resolution microscope N-SIM (Nikon), and were processed with NIS-Elements software (Nikon). For quantitative analysis of mitochondrial morphology, images were first thresholded and then converted to binary images by using ImageJ software[69,70]. Individual particle (mitochondria) were analyzed for mitochondrial area and number of fragments.

**Metabolome analysis**. Sort-purified $5 \times 10^4$ LSK cells from three mice were pooled, washed and then frozen. Metabolite extraction from sorted cells for metabolome analysis was performed as described previously[71]. Briefly, frozen sorted cell fractions together with internal standard (IS) compounds (2-morpholinoethanesulfonic acid (MES) and 1,3,5-benzenetricarboxylic acid (trimesate)) was suspended in ice-cold methanol (500 µL) followed by the addition of an equal volume of chloroform and 0.4 times the volume of ultrapure water (LC/MS grade, Wako). The suspension was then centrifuged at $15,000 \times g$ for 15 min at 4 °C. After centrifugation, the aqueous phase was filtered using an ultrafiltration tube (Ultrafree MC-PLHCC, Human Metabolome Technologies). The filtrate was concentrated with a vacuum concentrator (SpeedVac, Thermo Fisher Scientific). The concentrated filtrate was dissolved in 25 µL of ultrapure water and used for ion chromatography (IC)–MS analyses as described below. For metabolome analysis focused on TCA-cycle intermediates, anion metabolites were measured using an orbitrap-type MS (Q-Exactive focus, Thermo Fisher Scientific), connected to a high performance IC system (ICS-5000+, Thermo Fisher Scientific) that enables highly selective and sensitive metabolite quantification owing to the IC-separation and the Fourier Transfer MS principle[72]. The IC was equipped with an anion electrolytic suppressor (Thermo Scientific Dionex AERS 500, Thermo Fisher Scientific) to convert the potassium hydroxide gradient into pure water before the sample enters the mass spectrometer. The separation was performed using a Thermo Scientific Dionex IonPac AS11-HC, 4-µm particle size column. IC flow rate was 0.25 mL/min supplemented post-column with 0.18 mL/min makeup flow of MeOH. The potassium hydroxide gradient conditions for IC separation are as follows: from 1 to 100 mM (0–40 min), 100 mM (40–50 min), and 1 mM (50.1–60 min), at a column temperature of 30 °C. The Q Exactive focus mass spectrometer was operated under an ESI-negative mode for all detections. Full mass scan ($m/z$ 70−900) was used at a resolution of 70,000. The automatic gain control (AGC) target was set at $3 \times 10^6$ ions, and maximum ion injection time (IT) was 100 ms. Source ionization parameters were optimized with the spray voltage at 3 kV and other parameters were as follows: transfer temperature at 320 °C, S-Lens level at 50, heater temperature at 300 °C, Sheath gas at 36, and Aux gas at 10.

**RNA-seq analysis**. Total RNA was isolated from sorted LSK cells using RNeasy Micro-kit (QIAGEN) and Agilent TapeStation high sensitivity RNA ScreenTape or Bioanalyzer RNA 6000 Pico kit (Agilent) was used to check for RNA quality and quantity. RNA libraries for young mice were prepared using 15 ng total RNA with the QIAseq FX Single cell RNA library kit (QIAGEN) according to the manufacturer's protocol. In this protocol polyA-selected mRNA was converted in cDNA, and then enzymatic fragmentation (incubation time for "fragment size = 300 bp" was used) and library preparation were performed using 1 µg cDNA. RNA libraries for aged mice were prepared using 1 ng total RNA with the NEBNext Single Cell/Low Input RNA Library Prep kit (New England Biolabs). Briefly, cDNA synthesis and amplification were performed by template-switching PCR, and libraries were prepared using the amplified cDNA fragments. The quality and quantity of these libraries were checked using Agilent TapeStation D1000 or Agilent high sensitivity DNA kit (Agilent) and KAPA Library Quantification Kits (KAPA BioSystems)/real-time PCR Systems Step One Plus (Applied Biosystems). These libraries were

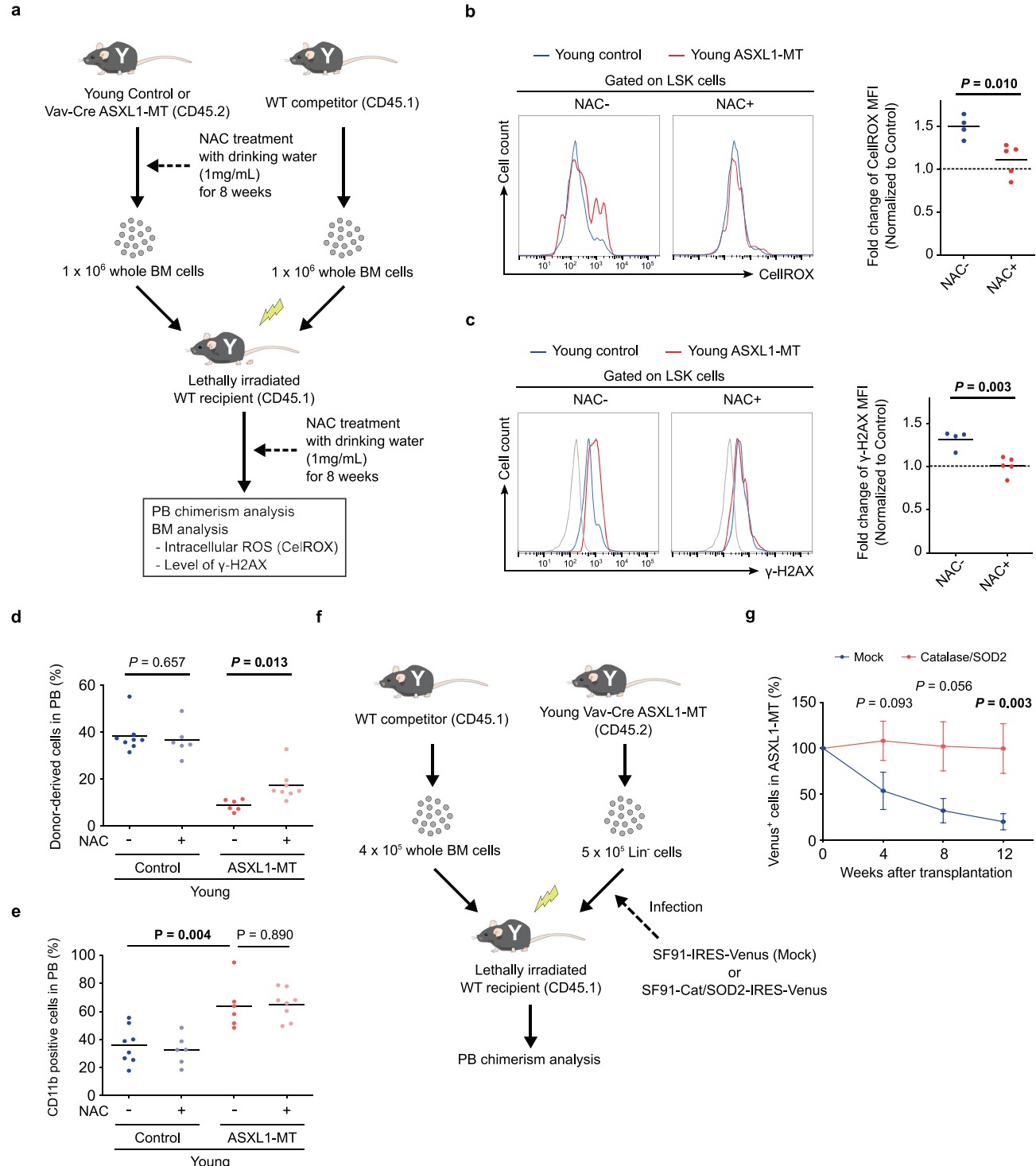

**Fig. 8 ROS-mediated DNA damage causes dysfunction of LT-HSCs. a** The experimental design for transplantation using whole bone marrow cells treated with an antioxidant scavenger NAC. Whole bone marrow cells isolated from young control and young *Vav-Cre* ASXL1-MT KI mice that had been treated with NAC for 8 weeks were transplanted into lethally irradiated recipient mice with competitor cells. Recipient mice were continuously treated with NAC after transplantation. **b**, **c** 6 weeks after transplantation, effects of NAC administration on ROS (**b**) and γ-H2AX levels (**c**) in LSK cells were determined (n = 4 (NAC−) and 5 (NAC+)). **d**, **e** The frequencies of donor-derived cells (**d**) and CD11b-positive cells in donor-derived cells (**e**) in peripheral blood were analyzed (n = 8 (Control-NAC−), 6 (Control-NAC+), 6 (ASXL1-MT-NAC−), and 8 (ASXL1-MT-NAC+)). **f** The experimental design for transplantations using Lin− cells transduced with ROS-detoxifying enzymes. Lin− cells from young *Vav-Cre* ASXL1-MT KI mice transduced with control or Catalase-SOD2 retrovirus were transplanted into lethally irradiated recipient mice. **g** Engraftment was assessed at the indicated weeks after transplantation (n = 7 (Mock) and 8 (Catalase-SOD2)). Data are mean ± s.e.m. Statistical significances are assessed by two-tailed Student's *t*-test. *P ≤ 0.05, **P ≤ 0.01.

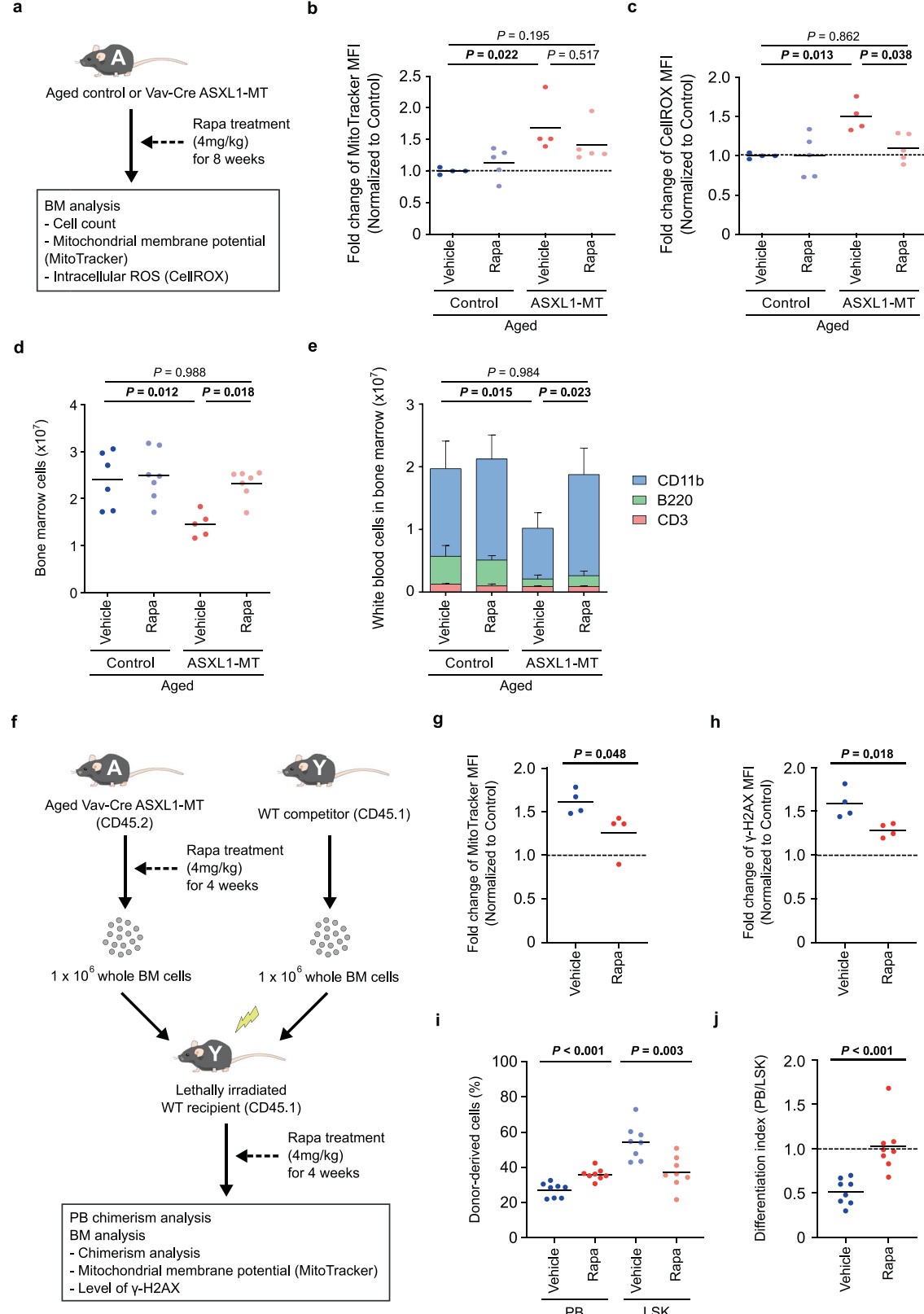

sequenced on the Illumina HiSeq2500 System with 2 × 100 nucleotide paired-end reads according to the manufacturer's protocol. Derived reads were processed using cutadapt (1.8.1) and fastx-toolkit (0.0.13) to remove Illumina adaptor sequence and to trim low-quality bases. Quality of reads was assessed using FastQC. Processed reads were aligned to GRCm38 reference transcripts using TopHat (2.1.1)—Cufflinks (2.1.1) pipeline to derive gene FPKM values[73–75].

Gene set enrichment analysis (GSEA) was performed using the GSEA tool from the Broad Institute (http://software.broadinstitute.org/gsea/). Heat maps were generated using heatmap.2 function of gplots package (3.0.4), from the Bioconductor. The microarray data for HSCs of young and aged wild-type mice are publically available at Gene Expression Omnibus (GEO) with the reference series tag "GSE 48893".

**Fig. 9 The dysregulated hematopoiesis in aged ASXL1-MT KI mice is reverted by treatment with rapamycin. a** Aged *Vav-Cre* ASXL1-MT KI mice and age-matched control mice were treated with rapamycin intraperitoneally every other day for 8 weeks. At the end of administrations, end-point analyses of bone marrow cells were conducted. **b, c** Mitochondrial membrane potential (**b**) and intracellular ROS levels (**c**) in LT-HSCs were analyzed ($n = 4$ (Control-Vehicle), 5 (Control-Rapamycin), 4 (ASXL1-MT-Vehicle), and 5 (ASXL1-MT-Rapamycin)). Results are expressed as fold change relative to vehicle control group. **d** Absolute numbers of bone marrow cells per leg ($n = 6$ (Control-Vehicle), 7 (Control-Rapamycin), 5 (ASXL1-MT-Vehicle), and 7 (ASXL1-MT-Rapamycin)). **e** Absolute numbers of white blood cells in bone marrow per leg ($n = 5$ (Control-Vehicle), 6 (Control-Rapamycin), 5 (ASXL1-MT-Vehicle), and 6 (ASXL1-MT-Rapamycin)). Data are mean ± s.d. **f** The experimental design for transplantation using whole bone marrow cells treated with rapamycin. Whole bone marrow cells isolated from aged *Vav-Cre* ASXL1-MT KI mice and age-matched control mice that had been treated with rapamycin for 4 weeks were transplanted into lethally irradiated recipient mice with competitor cells. Recipient mice were continuously treated with rapamycin after transplantation. **g, h** 4 weeks after transplantation, effects of rapamycin treatment on mitochondrial membrane potential (**g**) and γ-H2AX levels (**h**) in LSK cells were determined ($n = 4$). **i, j** 4 weeks after transplantation, the frequency of donor-derived cells in peripheral blood and LSK cells were analyzed ($n = 8$) (**i**). Differentiation index, which is obtained by dividing the frequency of donor-derived cells in peripheral blood by LSK cells, was calculated (**j**). Statistical significances are assessed by one-way ANOVA with Tukey–Kramer's post-hoc test (**b–e**) or two-tailed Student's *t*-test (**g–j**). *$P \leq 0.05$, **$P \leq 0.01$, ***$P \leq 0.001$.

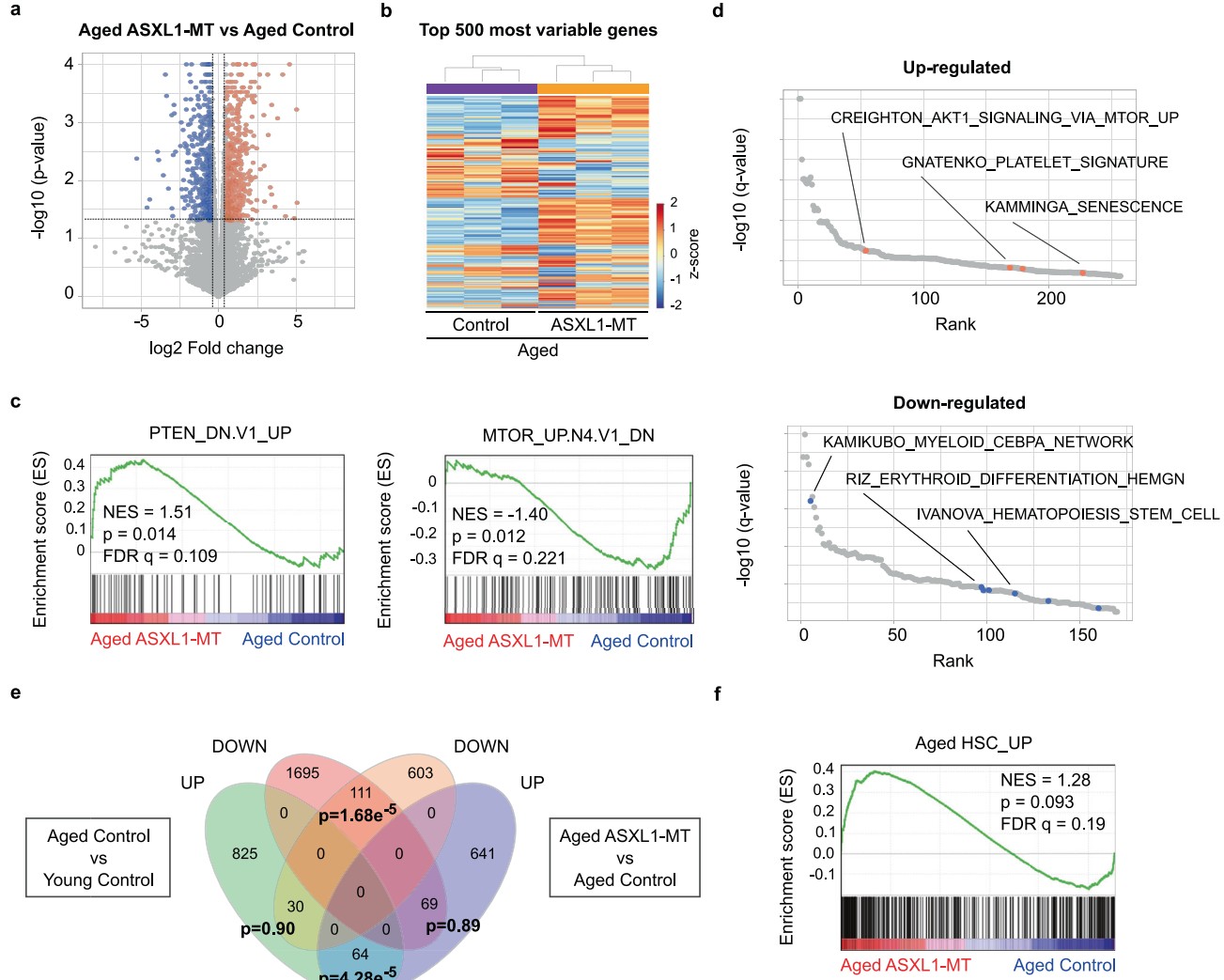

**Fig. 10 ASXL1-MT enforces age-associated gene expression patterns. a** Volcano plot displaying differentially expressed genes in LT-HSCs between aged *Vav-Cre* ASXL1-MT KI mice and aged-matched control mice ($n = 3$). **b** The heatmap of the top 500 most variable genes ($n = 3$). **c** GSEA using genesets up-regulated by PTEN deficiency (PTEN_DN.V1_UP) and negatively regulated by mTOR signaling (MTOR_UP.N4.V1_DN). **d** Up-regulated or down-regulated genesets (MSigDB C2_CGP: chemical and genetic perturbations) in LT-HSCs from aged *Vav-Cre* ASXL1-MT KI mice. Up-regulated genesets involved in Akt/mTOR signaling, platelet signature, and senescence signature are shown with orange circles. Down-regulated genesets involved in myeloid differenciation, erythroid differentiation, and HSC signature are shown with blue circles. **e** Venn diagrams showing up-regulated or down-regulated genes in LT-HSCs of (1) aged *Vav-Cre* ASXL1-MT KI mice compared to aged control mice and (2) aged wild-type mice compared to young wild-type mice (GSE48893). Statistical significances are assessed by two-sided Fisher's exact test. **f** GSEA using a geneset up-regulated in LT-HSCs with age (GSE48893).

**Comet assay**. Alkaline comet assays were performed with CometAssay Kit (Trevigen). Sort-purified LT-HSCs were embedded in LM Agarose and transferred onto Comet Slides. Cells were then lysed with lysis solution and soaked in alkaline solution (200 mM NaOH, 1 mM EDTA). Comet Slides were subjected to electrophoresis in alkaline solution followed by staining with SYBR Green. Analysis was performed using the ImageJ software.

**Single cell liquid culture**. Freshly isolated LT-HSCs were single cell sorted into 96-well round-bottom plates and were cultured in IMDM supplemented with 20% FBS, 1% penicillin/streptomycin, 2 mM L-glutamine, 50 mM 2-Mercaptoethanol (Thermo Fisher Scientific; 21985-023), SCF (50 ng/mL), TPO (50 ng/mL), IL-3 (10 ng/mL), IL-6 (10 ng/mL) for 2 weeks.

**Subcellular fractionation**. To isolate the nucleus, cells were washed with ice-cold PBS, harvested with hypotonic buffer (10 mM HEPES pH 7.0, 1.5 mM $MgCl_2$, 10 mM KCl, 0.5 mM DTT, 0.1% TritonX-100, 2 mM $Na_3VO_4$, 2 mM PMSF, 50 mM NaF, 10 μM MG132) supplemented with proteinase inhibitor (cOmplete Mini, Roche, 11836153001), and subsequently incubated for 20 min on ice. Lysates were centrifuged at 3500 × $g$ for 5 min and subsequently separated into supernatants (cytoplasmic fraction) and pellets (nuclear fraction). The pellets were lysed with lysis buffer (TBS, 0.5% NP-40, 2 mM $Na_3VO_4$, 2 mM PMSF, 50 mM NaF, 10 μM MG132, proteinase inhibitor cocktail) followed by incubation for 30 min on ice with pipetting occasionally. Lysates were then centrifuged at 11,000 × $g$ for 30 min at 4 °C and supernatants were collected as a nuclear fraction.

**Coimmunoprecipitation and western blotting**. 293T cells were transfected with plasmids using PEI. 48 h after transfection, cells were washed with ice-cold PBS and were lysed with cell lysis buffer (50 mM Tris-HCl, 1 mM EDTA, 150 mM NaCl, 1% TritonX-100, 0.1% SDS, 2 mM $Na_3VO_4$, 2 mM PMSF, 50 mM NaF, 10 μM MG132) supplemented with proteinase inhibitor (cOmplete Mini, Roche, 11836153001). Lysates are immunoprecipitated with anti-FLAG (Sigma, F1804, 1:100) or anti-AKT1/2/3 (Santa Cruz Biotechnology, sc-81434, 1:20) antibody using Dynabeads Protein G (Thermo Fisher Scientific, 10004D) for 10 min at room temperature. Immunoprecipitated samples were subjected to electrophoresis, transfer to nitrocellulose membranes, and blocking with 5% bovine serum albumin (BSA)/TBS for 1 h at room temperature. Samples were then incubated with primary antibodies in 1% BSA/0.1% Tween 20/TBS overnight at 4 °C, washed and incubated with secondary antibodies in 1% BSA/0.1% Tween 20/TBS for 1 h at room temperature. The primary antibodies used in immunoblotting are as follows: anti-AKT1/2/3 (Santa Cruz Biotechnology, sc-81434, 1:200), anti-Akt (Pan) (Cell Signaling Technology, #4691, 1:1000), anti-Phospho-Akt (Ser473) (Cell signaling Technology, #4060, 1:2000), anti-BAP1 (Santa Cruz Biotechnology, sc-28383, 1:200), anti-FLAG (Sigma, F1804, 1:1000), anti-GAPDH (Cell signaling Technology, #5174, 1:2000), anti-Myc (Roche, #11814150001, 1:1000), anti-HA (Roche, #11867423001, 1:1000), anti-α-Tubulin (SIGMA, T9026, 1:3000), and anti-Lamin B1 (Santa Cruz Biotechnology, sc-374015, 1:50).

**Human data analysis**. The clinical microarray data for CD34-positive bone marrow cells from 159 MDS patients, including 13 patients with *DNMT3A* mutations, 33 patients with *TET2* mutations, and 21 patients with *ASXL1* mutations, and 17 healthy controls are publically available at Gene Expression Omnibus (GEO) with the reference series tag "GSE58831".

**Statistics and reproducibility**. Data are expressed as mean ± standard deviation (s.d.) or standard error of mean (s.e.m.). $P$ values were calculated using unpaired, two-tailed Student's $t$-test, Mann–Whitney rank sum test, one-way ANOVA with Tukey–Kramer's post-hoc test, or two-sided Fisher's exact test as indicated in figure legend. The significance between the survival of cohorts was assessed by Kaplan–Meier survival analysis and log-ranked tests. $P$ values < 0.05 were considered significant. All calculations were performed by using GraphPad Prism software. Sample size '$n$' indicates biological replicates. No randomization or blinding was used and no animals were excluded from the analysis.

**Reporting summary**. Further information on research design is available in the Nature Research Reporting Summary linked to this article.

## Data availability

RNA-seq data are available in sequence read archive (SRA) database (accession ID: PRJNA673672). Metabolome analysis data are available in MassIVE (accession ID: MSV000086816). The datasets generated or analyzed during the current study are available from the corresponding author (kitamura@ims.u-tokyo.ac.jp) on reasonable request. Source data are provided with this paper.

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

## Acknowledgements

We thank K. Matsuda for sharing *p53*$^{-/-}$ mice, Michael D. Milsom for providing SF91-IRES-GFP and SF91-Cat/SOD2-IRES-GFP vectors, IMSUT FACS Core Laboratory for flow cytometry and Nikon Imaging Laboratory for microscopic analysis. This work was supported by Grant-in-Aid for Scientific Research (B) (No. 15H04855) (to T.K.), Grant-in-Aid for Scientific Research on Innovative Areas "Stem Cell Aging and Disease" (No. 17H05634), The Tokyo Biochemical Research Foundation (to T.K.), Japanese Society of Hematology (to T.K.), The Japan foundation for Aging and Health (to S.G.), Suzuken Memorial Foundation (to S.G.). Infrastructure of metabolomics was partly supported by JST ERATO Suematsu Gas Biology (2010–2015) (to Y.S. and M.S.). O.A.-W. is supported by grants from a Leukemia & Lymphoma Society Specialized Center for Research award as well as funding from the Edward P. Evans MDS Foundation and the Henry & Marilun Taub Foundation.

## Author contributions

Ta.F. designed and performed experiments and analyzed data with assistance from S.A., A.T., S.S., N.S., To.F., Y.T., Ts.F, R.T., and K.Y. Y.S., and M.S. performed metabolome analysis. S.Y., A.M., K.Y., Y.I., Y.F., and T.S. performed RNA-seq analysis. D.I., H.M., E.K.N., and O.A.-W. provided critical materials, knowledge on their use, and concepts. H.H. generated ASXL1-MT cKI mice. S.G. and T.K. guided research and wrote the paper with Ta.F.

## Competing interests

O.A.-W. has served as a consultant for H3B Biomedicine, Foundation Medicine Inc., Merck, and Janssen and serves on the scientific advisory board for Envisagenics Inc.; O.A.-W. has received personal speaking fees from Daiichi Sankyo. Other authors declare no competing interests.
