## [Peer Review File · Nature Communications]

Reviewers' comments:

Reviewer #1 (Remarks to the Author): <The same comments also attached - in PDF with formatting of the comments>

NCOMMS-19-39409-T, Prof Kitamura and co-workers, Mutant ASXL1 induces clonal hematopoiesis through activation of Akt/mTOR pathway

The study by Fujino et al. provides an exhaustive description of the cellular phenotypes of ASXL1-mutant (MT) knock-in mice, extending the information presented in a previous study (Nagase et al., Ref 25). These mice express a gain-of-function truncation mutant of ASXL1 specifically in hematopoietic cells, which confers a striking increase in the catalytic activity of BAP1, the deubiquitinase that is the obligate partner protein of ASXL1. The authors confirm their previous finding that hematopoietic stem/ precursor cells (HSPCs) from young ASXL1-MT mutant mice display a competitive disadvantage in mixed transplantation assays. Unexpectedly, aged ASXL1-MT mutant mice accumulate more LSK cells (HSPCs) and more long-term hematopoietic stem cells (LT-HSC), which however show biased lineage specification and lack the ability to give rise to all hematopoietic cell types. The mutant mice eventually show clonal hematopoiesis, a feature also observed in human patients with similar truncation mutations of ASXL1.

The authors propose the interesting hypothesis that the mechanism of clonal hematopoiesis in ASXL1-MT patients and mice is very different from that observed for DNMT3A or TET2 mutations. Their explanation rests on the observation of a mild increase in AKT and mTOR activation in several hematopoietic cell types from ASXL1-MT mice. They show that treatment with the mTORC1 inhibitor rapamycin reduces the number of LSK cells and LT-HSC in aged mice, and propose a mechanism that involves deubiquitylation and consequent stabilization of phosphorylated AKT by the ASXL1-MT/ BAP1 complex. The ASXL1-MT HSPCs also show increased mitochondrial membrane potential and increased oxidative phosphorylation, in parallel with higher levels of reactive oxygen species (ROS) and increased DNA double-strand breaks (DSBs), as judged by increased staining with phosphorylated H2A.X (gamma-H2A.X).

Overall, the study is carefully performed and contains several interesting observations – e.g. it briefly mentions the phenotype of ASXL1-MT/ p53 KO animals, presents data on the accumulation of gamma-H2AX in aged ASXL1-MT HSPCs, and provides a provocative explanation for the proliferative advantage conferred by ASXL1-MT – i.e. continuous activation of the AKT/ mTOR pathway. As such it is an important contribution, that would nevertheless be considerably strengthened by attention to the following major and minor points.

Major comments:

1. To avoid confusion, the authors need to state in every relevant figure whether the cells being used were from young or old WT versus ASXL1-MT mice and exactly what experiment was performed. Ideally they would provide flow-charts for each experiment and labels over the figure panels to distinguish young from old (as in Figures 4c, right and 4d). Also they should indicate what age range they mean in each case – 6-12 weeks for young mice and ~70 weeks (as specified in Nagase et al.) for old mice?
2. In the cell cycle assay of Figure 1h, the authors show a lower frequency of LT-HSC in the G0 phase, but a higher frequency of LT-HSCs in S/G2/M phase, in young ASXL1-MT-KI mice. This finding suggests that LT-HSC in these mice lose quiescence and are in an actively dividing state (although they also undergo more apoptosis). A side-by-side comparison of young and old ASXL1-MT-KI HSPCs against the corresponding control HSPCs would be appropriate and useful here. Please also describe how the cell cycle analysis was performed – BrdU and 7-AAD staining?

3. Similar comments apply to Figures 4 and 6. In Figures 4a-c, the authors use flow cytometry and GSEA of RNA-seq data to show that LSK cells from young ASXL1-MT-KI mice have increased Akt/mTOR signaling; is this also true for LSK cells from aged ASXL1-MT-KI mice? Similarly for Figures 6a-d, were the experiments performed in young or aged ASXL1-MT-KI HSPCs? If at all possible, it would be useful to see a comparison of HSPCs from young and old mice in both figures.
4. In figure 5a-d, the authors show that AKT co-immunoprecipitates with both ASXL1 and BAP1. Does the increase in phospho-AKT directly or indirectly affect the interaction between ASXL1 and BAP1, which stabilizes both partner proteins?
5. Although BAP1 most notably opposes histone H2A monoubiquitination, BAP1 is one of the two ubiquitin C-terminal hydrolase family members that is capable of disassembly of specifically K48-linked polyubiquitin chains.¹ Akt has also been reported to be modified by K63-linked polyubiquitin, in addition to the K48-linked polyubiquitin chains that target proteins for proteasomal degradation.²⁻⁴ Accordingly, it was shown that Akt K63-linked ubiquitination affects its signaling, but not stability.^{2,5} Could the authors provide evidence that the observed changes in Akt ubiquitin levels with ASXL1-MT are due to K48-linked ubiquitin? An experiment showing that the effect of ASXL1-MT on Akt ubiquitin levels is lost with mutant K48R-ubiquitin would strengthen this conclusion.
6. BAP1 exhibits predominantly nuclear localization⁶, containing at least one functional NLS⁷; whereas, Akt is activated at PI-(3,4,5)-P3 and PI-(3,4)-P2-containing membranes. It has been shown that activated and ubiquitinated Akt is more efficiently trafficked into the nucleus³; however, more recent work has suggested that active Akt is predominantly associated with cellular membranes.⁸ Are the authors proposing that the interaction between ASXL1-MT and Akt occurs in the nucleus, and if so, can they provide evidence of the interaction?
7. The authors show in Figures 7c and 7d that treatment with rapamycin rescued the number of differentiated white blood cells and bone marrow cellularity in aged ASXL1-MT-KI mice. Supplementary Figure 5 shows that mTOR inhibition or Akt inhibition both recover the elevated mitochondrial membrane potential and ROS. Similar experiments performed with an Akt inhibitor would strengthen the authors' conclusions that ASXL1-MT-mediated deubiquitination and stabilization of phosphorylated Akt is responsible for the LT-HSC expansion.

Minor Comments:

1. Figures 3c, d: please perform the appropriate statistical tests and report p-values in the graphs.
2. Figures 4B, 4C, and 5H reference phosphorylated Akt without specifying the site of phosphorylation. The specific phosphorylation site assessed in each experiment should be listed in the figure or figure legends (e.g. p-AKT (Ser473)). Also, for the flow cytometry of Figure 4b (p-AKT and pS6 staining), it would be preferable to use an anti- isotype antibody as the negative control.
3. Several of the co-immunoprecipitation experiments in Figure 5 propose an interaction between Akt and ASXL1-MT without the use of a control antibody. The authors should show a representative experiment including an IgG control to demonstrate that the co-IP is specific and does not represent the very common artefact of non-specific protein adsorption to beads.
4. In Figures 5a-f, the experiments involve overexpression in 293 T cells. Can the authors also perform the experiments in bone marrow cells, using antibodies to either the endogenous proteins or to ectopically-expressed epitope-tagged proteins?
5. In Figure 6b, the authors should explain the meaning behind the color scale in their heatmap (Z-score?).
6. In Supplementary Figure 4a, the authors may want to add the frequency of the LSK population in p53 KO mice for reference.
7. RNA-seq of HSPC from young ASXL1-MT-KI mice revealed upregulation of genes involved in observed mild activation of AKT/mTOR and oxidative phosphorylation. Does the RNA-seq reveal anything significant about the expression of glycolytic pathway genes in young and old mice?
8. In figure 7, the authors detected mitochondrial membrane potential and ROS after Rapamycin treatment, is it possible to detect the DNA damage, for example gammaH2A.X staining.

References

1. Scheuermann JC, de Ayala Alonso AG, Oktaba K, Ly-Hartig N, McGinty RK, et al. Histone H2A deubiquitinase activity of the Polycomb repressive complex PR-DUB. *Nature*. 2010;465(7295):243–247.
2. Yang W-L, Wang J, Chan C-H, Lee S-W, Campos AD, et al. The E3 Ligase TRAF6 Regulates Akt Ubiquitination and Activation. *Science* (80-.). 2009;325(5944):1134–1138.
3. Fan C-D, Lum MA, Xu C, Black JD, Wang X. Ubiquitin-dependent regulation of phospho-AKT dynamics by the ubiquitin E3 ligase, NEDD4-1, in the insulin-like growth factor-1 response. *J. Biol. Chem*. 2013;288(3):1674–84.
4. Suizu F, Hiramuki Y, Okumura F, Matsuda M, Okumura AJ, et al. The E3 Ligase TTC3 Facilitates Ubiquitination and Degradation of Phosphorylated Akt. *Dev. Cell*. 2009;17(6):800–810.
5. Yang W-L, Wu C-Y, Wu J, Lin H-K. Regulation of Akt signaling activation by ubiquitination. *Cell Cycle*. 2010;9(3):487–97.
6. Jensen DE, Proctor M, Marquis ST, Gardner HP, Ha SI, et al. BAP1: A novel ubiquitin hydrolase which binds to the BRCA1 RING finger and enhances BRCA1-mediated cell growth suppression. *Oncogene*. 1998;16(9):1097–1112.
7. Ventii KH, Devi NS, Friedrich KL, Chernova TA, Tighiouart M, et al. BRCA1-associated protein-1 is a tumor suppressor that requires deubiquitinating activity and nuclear localization. *Cancer Res*. 2008;68(17):6953–6962.
8. Ebner M, Lučić I, Leonard TA, Yudushkin I. PI(3,4,5)P3 Engagement Restricts Akt Activity to Cellular Membranes. *Mol. Cell*. 2017;65(3):416-431.e6.

Reviewer #2 (Remarks to the Author):

Age-related clonal hematopoiesis (ARCH) occurs when a single mutant hematopoietic stem cell (HSC) contributes to a significant clonal proportion of mature blood lineages. ARCH is associated with increased risks of de novo and therapy-related hematological neoplasms, suggesting that mutations identified in ARCH likely drive disease development. Mutations most commonly detected in CH include the epigenetic regulators DNMT3A, TET2, and ASXL1. While loss of DNMT3A or TET2 enhances HSC self-renewal, loss of ASXL1 and expression of mutant ASXL1 decrease HSC functions. The mechanisms by which mutant ASXL1 drives clonal hematopoiesis are largely unknown.

The authors examined the influence of mutant ASXL1 on physiological HSC aging using the knockin (KI) mice expressing a C-terminally truncated form of ASXL1-mutant in the hematopoietic system (ASXL1-MT-VavCre⁺ mice). They found that aged ASXL1 mutant HSCs show decreased repopulating potential in transplantation assays. RNA-seq assays revealed that Akt/mTOR pathway is activated in young ASXL1 mutant HSCs. They further showed that mutant ASXL1 interacts with BAP, leading to deubiquitination and activation of AKT. Interestingly, pharmacologically inhibition of the mTOR pathway ameliorates aberrant expansion of the HSC compartment as well as dysregulated hematopoiesis in aged ASXL1-MT-KI mice. Thus, the authors demonstrated that mutant ASXL1 promotes HSC aging through activating the Akt/mTOR pathway.

In addition to HSC aging, the authors utilized ASXL1-MT-Mx1Cre⁺ mice to investigate the impact of mutant ASXL1 on clonal hematopoiesis. They treated ASXL1-MT-Mx1Cre⁺ mice with pI:pC to partially induce mutant ASXL1 expression in hematopoietic system and found that HSCs expressing mutant ASXL1 acquire clonal advantage during aging.

Overall, this is an interesting study that provides a novel mechanism by which mutant ASXL1 promotes HAS aging. However, studies on CH were descriptive and data were not convincing. The mechanistic link between the impact of mutant ASXL1 on HSC aging and CH is not clear. A significant

amount of experiments will be needed to strengthen the manuscript and support conclusion drawn.

Major Criticisms/Suggestions:

1. Both young and aged mutant HSCs show decreased repopulating potential in transplantation assays as shown in Figures 1 and 2, which may be due to impaired HSC homing capability. Therefore, authors should perform homing assays using both young and aged mutant HSCs.
2. Given that the paper focuses on determining the impact of mutant ASXL1 on HSCs during aging, RNA-seq studies using aged HSCs are needed to elucidate the molecular mechanisms by which mutant ASXL1 promotes HSC aging.
3. Figure 4b showed that young ASXL1 mutant HSPCs show increased levels of pAKT and pS6. What about the levels of pAKT and pS6 in aged mutant HSPCs?
4. Hematopoiesis varies between mice, especially during aging. However, all data from Figure 3 were generated from only two aged ASXL1-MT-Mx1Cre⁺ mice (n=2). Given that the conclusion of the paper is mainly based on data from Figure 3, more aged mutant mice, at least 8 to 10 mice, are needed to generate convincing and solid data.
5. Figure 3A. The authors treated ASXL1-MT-Mx1Cre⁺ mice with pI:pC (250ug, IP) every other day for three times to induce mutant ASXL1 expression in hematopoietic cells. This treatment should induce GFP expression in majority of hematopoietic cells. What are the percentage of GFP⁺ cells following pIpC treatment?
6. pI:pC treatment mimics interferon response. Hematopoietic aging is manifested by chronic inflammation. There are increased levels of pro-inflammatory cytokines such as interferon γ (IFN γ) in both peripheral blood serum and bone marrow fluid of aged mice. The authors found that mutant HSCs (GFP⁺) expand in ASXL1-MT-Mx1Cre⁺ mice with age (Figure 3A). The increase in GFP⁺ cells with age may not be due to increased clonal hematopoiesis but increased secretion of IFN γ in mutant ASXL1 mice, inducing more mutant ASXL1 expression in hematopoietic cells. Thus, the authors should examine the levels of IFN γ and other pro-inflammatory cytokines in PB serum and bone marrow fluid of aged WT and ASXL1-MT-Mx1Cre⁺ mice with age

Reviewer #3 (Remarks to the Author):

Comments from the referee:

In the first part of the present study, the authors reproduced previous published results by comparing the hematopoiesis of young and old mice in figures 1 and 2 (increased HSC frequency, decrease competitive engraftment, increased cell cycle and apoptosis) and show evidence that HSC-CD41⁺ are increased but not functional. In contrast, after crossbreeding their KI mice with Mx1-cre mice, they show that the expression of ASXL1-mut in native hematopoiesis leads, this time, to increased clonal advantage.

In the second part of the study, they investigate the mechanism by which ASXL1-mut induces growth advantage of pLT-HSC but alters their function. This is the most interesting and novel part. Using biochemical and genetic studies, they found that ASXL1-mut/BAP1 promotes AKT deubiquitination and increases its phosphorylation. This leads to an increase in mitochondrial activity mainly through mTOR, in ROS levels and in DNA damages in a P53-dependent manner. Manipulations of this pathway (ROS inhibition with NAC, overexpressing catalase or AKT/mTOR inhibition) rescued ASXL1-mut clonal disadvantage in treated mice. These experiments indicate that the AKT/mTOR pathway is involved in the cell cycle of HSC but also alters their DNA and triggers apoptosis resulting in an overall increased pool of non-functional HSC.

Finally, they checked AKT and mTOR signaling in public transcriptome of MDS patients and found

enrichment suggesting that it is relevant for the pathology, making this pathway a good target to inhibit in patients with ASXL1-mut.

This is a very interesting study that helps to better understand the development of hematological malignancies but there are several aspects that need improvement or clarification.

1) In contrast to study performed with Vav-Cre mice, authors show that the expression of ASXL1-mut phenotype in native hematopoiesis, after crossbreeding with Mx1-cre mice, leads to an increased clonal advantage. This difference is explained by a deleterious effect of transplantation in ASXL1-mut-mediated clonal advantage. I found these experiments to be a bit confusing for the understanding of the paper since all the next experiments are performed with vav-Cre. Moreover, since Cre, under Mx1 promoter control, can be expressed in non-hematopoietic cells, another explanation could be that the other cells or microenvironment may play a role in this difference. What happens in term of clonal advantage in the PB if BM cells from Mx1-Cre/ASXL1-mut mice are engrafted in WT mice or conversely if Vav-Cre/ASXL1-mut BM cells are engrafted in Mx1-Cre/ASXL1-mut after induction? Alternatively, ASXL1-mut-induced competitive advantage could be due to aging of microenvironment in addition to aging of hematopoiesis. In this regard, it would be informative to perform transplantation of Vav-Cre/ASXL1-mut BM cells in aged recipients to assay clonal advantage.

2) In their study, the authors check the mechanism underlying the growth advantage of pLT-HSC expressing ASXL1-mut and leading to their clonal disadvantage. Thus, the title seems overstated since the mut-ASXL1 did not induce "clonal hematopoiesis" in the Vav-Cre mouse model.

3) The finding that AKT is overactivated in LSK is very interesting.

Are the results of flow cytometry obtained in steady state? Since sometimes the phosphorylated form of intracellular protein are difficult to detect, the authors should show the histogram plots for Ig control antibodies and, if possible, a positive control (in vitro treatment with SCF) and/or a negative control (in vitro treatment with PI3K inhibitor).

4) The demonstration that ASXL1-mut/BAP1 promotes AKT deubiquitination is novel and of great interest. Why did the authors choose to focus on AKT2 and not AKT1? Are the findings also true with AKT1?

5) Moreover, since the ASXL1-mut/BAP1 promotes AKT deubiquitination and stabilization, is total AKT increased in LSK? In 293T cells, is AKT half-life affected by ASXL1-mut? Maybe cycloheximide experiments should be conducted?

6) The experiment performed on P53-/-ASXL1-mut are interesting to understand the mechanism of action. Did the authors check the HSC compartment in these mice? Since p53 impacts genetic instability and ROS level through transcriptional mechanism, it could be informative to measure ROS levels and DNA damages in these mice.

Minor

Authors should verify the indicated number of mice in figure legends. For instance, in figure 1 (h) there seem to be 5 mice in the figure and not 6.

In addition, authors should detail in figure legends the mouse model that was used for each experiment: Vav-Cre or Mx1-Cre.

Reviewers' comments:

Reviewer #1 (Remarks to the Author): <The same comments also attached - in PDF with formatting of the comments>

NCOMMS-19-39409-T, Prof Kitamura and co-workers, Mutant ASXL1 induces clonal hematopoiesis through activation of Akt/mTOR pathway

The study by Fujino et al. provides an exhaustive description of the cellular phenotypes of ASXL1-mutant (MT) knock-in mice, extending the information presented in a previous study (Nagase et al., Ref 25). These mice express a gain-of-function truncation mutant of ASXL1 specifically in hematopoietic cells, which confers a striking increase in the catalytic activity of BAP1, the deubiquitinase that is the obligate partner protein of ASXL1. The authors confirm their previous finding that hematopoietic stem/ precursor cells (HSPCs) from young ASXL1-MT mutant mice display a competitive disadvantage in mixed transplantation assays. Unexpectedly, aged ASXL1-MT mutant mice accumulate more LSK cells (HSPCs) and more long-term hematopoietic stem cells (LT-HSC), which however show biased lineage specification and lack the ability to give rise to all hematopoietic cell types. The mutant mice eventually show clonal hematopoiesis, a feature also observed in human patients with similar truncation mutations of ASXL1.

The authors propose the interesting hypothesis that the mechanism of clonal hematopoiesis in ASXL1-MT patients and mice is very different from that observed for DNMT3A or TET2 mutations. Their explanation rests on the observation of a mild increase in AKT and mTOR activation in several hematopoietic cell types from ASXL1-MT mice. They show that treatment with the mTORC1 inhibitor rapamycin reduces the number of LSK cells and LT-HSC in aged mice, and propose a mechanism that involves deubiquitylation and consequent stabilization of phosphorylated AKT by the ASXL1-MT/ BAP1 complex. The ASXL1-MT HSPCs also show increased mitochondrial membrane potential and increased oxidative phosphorylation, in parallel with higher levels of reactive oxygen species (ROS) and increased DNA double-strand breaks (DSBs), as judged by increased staining with phosphorylated H2A.X (gamma-H2A.X).

Overall, the study is carefully performed and contains several interesting observations – e.g. it briefly mentions the phenotype of ASXL1-MT/ p53 KO animals, presents data on the accumulation of gamma-H2AX in aged ASXL1-MT HSPCs, and provides a provocative explanation for the proliferative advantage conferred by ASXL1-MT – i.e. continuous activation

of the AKT/ mTOR pathway. As such it is an important contribution, that would nevertheless be considerably strengthened by attention to the following major and minor points.

Thank you for your thoughtful comments. We could improve our manuscript thanks to your valuable advice.

Major comments:

1. To avoid confusion, the authors need to state in every relevant figure whether the cells being used were from young or old WT versus ASXL1-MT mice and exactly what experiment was performed. Ideally they would provide flow-charts for each experiment and labels over the figure panels to distinguish young from old (as in Figures 4c, right and 4d). Also they should indicate what age range they mean in each case – 6-12 weeks for young mice and ~70 weeks (as specified in Nagase et al.) for old mice?

Thank you very much for raising these points. To clarify them, we have now described the derivation of the cells from young or aged mice in every relevant figure. We also indicated the range of the ages for young and aged mice in the text (page6 line116, page7 line140, and “Mice” in the Methods section). In addition, we have now added flow-charts for each experiment and labels over the figure panels, according to the reviewer’s comment.

2. In the cell cycle assay of Figure 1h, the authors show a lower frequency of LT-HSC in the G0 phase, but a higher frequency of LT-HSCs in S/G2/M phase, in young ASXL1-MT KI mice. This finding suggests that LT-HSC in these mice lose quiescence and are in an actively dividing state (although they also undergo more apoptosis). A side-by-side comparison of young and old ASXL1-MT KI HSPCs against the corresponding control HSPCs would be appropriate and useful here. Please also describe how the cell cycle analysis was performed – BrdU and 7-AAD staining?

According to the reviewer’s comment, we have added the data of young control mice to Fig 5d (Fig. 4e in the original manuscript) and made it possible to compare young and aged ASXL1-MT KI HSPCs against young and aged control HSPCs using young control mice as a basis. We have also explained how cell cycle analyses were performed in every relevant figure and “Cell cycle analysis” in the method section.

3. Similar comments apply to Figures 4 and 6. In Figures 4a-c, the authors use flow cytometry and GSEA of RNA-seq data to show that LSK cells from young ASXL1-MT KI mice have

increased Akt/mTOR signaling; is this also true for LSK cells from aged ASXL1-MT KI mice? Similarly for Figures 6a-d, were the experiments performed in young or aged ASXL1-MT KI HSPCs? If at all possible, it would be useful to see a comparison of HSPCs from young and old mice in both figures.

(Activity of Akt/mTOR pathway)

Thank you very much for the important comments. According to the reviewer's comments, we have performed additional experiments to examine activities of Akt/mTOR signaling in young and aged *Vav-Cre* ASXL1-MT KI mice. Flow cytometry analyses demonstrated activation of Akt/mTOR signaling in LT-HSCs of aged *Vav-Cre* ASXL1-MT KI mice to the same degree in young *Vav-Cre* ASXL1-MT KI mice compared with age-matched control mice. RNA-seq analyses of aged LT-HSCs also revealed activation of Akt/mTOR pathway in *Vav-Cre* ASXL1-MT KI mice. We have added the results in Fig. 4d, 4e (flow cytometry) and Fig. 10c (RNA-seq). The text has been changed accordingly.

(Mitochondrial membrane potential and ROS levels)

We thank the reviewer for pointing this out. For young mice, mitochondrial membrane potentials and ROS levels are displayed as Fig. 7b and Fig. 7f (Fig. 6c and 6d in the original manuscript), respectively. For aged mice, these are displayed as Fig. 9b and 9c (Fig. 7a and 7b in the original manuscript), respectively. When compared to the young mice, mitochondrial membrane potentials and ROS levels were lower (see attached figure). These findings in aged LT-HSCs has been reported (Theodore et al, *Nature*, 2017), but if we present the control of young mice in the old mouse results in Fig. 9b and 9c as shown in the attached figure, the readers may be confused. We will abide by the decision of the reviewer and the editor.

4. In figure 5a-d, the authors show that AKT co-immunoprecipitates with both ASXL1 and BAP1. Does the increase in phospho-AKT directly or indirectly affect the interaction between ASXL1 and BAP1, which stabilizes both partner proteins?

In response to the reviewer's comment, we used Akt1 AA and DD mutants, which are phosphorylation-resistant and mimic mutants of Akt1 respectively, to investigate if phosphorylation of Akt1 affect the interaction between ASXL1 and BAP1. However, it did not change. In this research, because we would like to argue that ASXL1-MT/BAP1 complex affects the stability of phosphorylated Akt, but not vice versa, we do not think that we need to add this result in the revised manuscript, but would like to show this to the reviewer (see attached figure).

5. Although BAP1 most notably opposes histone H2A monoubiquitination, BAP1 is one of the two ubiquitin C-terminal hydrolase family members that is capable of disassembly of specifically K48-linked polyubiquitin chains.¹ Akt has also been reported to be modified by K63-linked polyubiquitin, in addition to the K48-linked polyubiquitin chains that target proteins for proteosomal degradation.^{2–4} Accordingly, it was shown that Akt K63-linked ubiquitination affects its signaling, but not stability.^{2,5} Could the authors provide evidence that the observed changes in Akt ubiquitin levels with ASXL1-MT are due to K48-linked ubiquitin? An experiment showing that the effect of ASXL1-MT on Akt ubiquitin levels is lost with mutant K48R-ubiquitin would strengthen this conclusion.

Thank you very much for raising this point. In response to the reviewer's comment, we have performed the experiments using K48R ubiquitin and K63R ubiquitin and could show that the effect of ASXL1-MT/BAP1 expression on Akt ubiquitination was lost when we used the K48R mutant but not K63R mutant, demonstrating that ASXL1-MT/BAP1 complex mainly deubiquitinates K48-linked ubiquitin. We have added this data to supplementary Figure 7b and have changed the text and figure legend accordingly.

6. BAP1 exhibits predominantly nuclear localization, containing at least one functional NLS; whereas, Akt is activated at PI-(3,4,5)-P3 and PI-(3,4)-P2-containing membranes. It has been shown that activated and ubiquitinated Akt is more efficiently trafficked into the nucleus³; however, more recent work has suggested that active Akt is predominantly associated with cellular membranes.⁸ Are the authors proposing that the interaction between ASXL1-MT and Akt occurs in the nucleus, and if so, can they provide evidence of the interaction?

Thank you very much for the important comment. In response to reviewer's comment, we performed IP-western using cytoplasmic and nuclear fractions. Interaction between ASXL1-WT/MT and Akt was detected in both fractions. Considering the localization of BAP1, ASXL1-MT/BAP1 complex could deubiquitinate phosphorylated Akt in the nucleus. On the other hand, we observed a clear interaction between ASXL1-MT and BAP1 in the cytoplasm. It is possible that cytoplasmic activities of ASXL1-MT/BAP1 complex is more important to activate Akt/mTOR pathway as components of this pathway are localized in the cytoplasm. We have added new results as a supplementary Figure 7a, and changed the text and figure legend accordingly.

7. The authors show in Figures 7c and 7d that treatment with rapamycin rescued the number of differentiated white blood cells and bone marrow cellularity in aged ASXL1-MT KI mice.

Supplementary Figure 5 shows that mTOR inhibition or Akt inhibition both recover the elevated mitochondrial membrane potential and ROS. Similar experiments performed with an Akt inhibitor would strengthen the authors' conclusions that ASXL1-MT-mediated deubiquitination and stabilization of phosphorylated Akt is responsible for the LT-HSC expansion.

Thank you very much for this valuable advice. Because it is currently impossible to secure many aged control and ASXL1-MT KI mice, we chose to do the experiment where an Akt inhibitor perifosine was singly administrated to five aged control and *Vav-Cre* ASXL1-MT-KI mice. From this small experiment, we could still find that Akt inhibition reduced the elevated mitochondrial membrane potential in aged *Vav-Cre* ASXL1-MT KI mice (Supplementary Fig. 11d and 11e). We also found that the decreased frequency of G0 cells in aged *Vav-Cre* ASXL1-MT KI mice were normalized to that of age-matched control mice (Fig. 5f and 5g). These results further confirm that ASXL1-MT-mediated Akt activation is responsible for the expansion of LT-HSC, and are now shown as Fig. 5f, 5g and Supplementary Fig. 11d, 11e. The text has been modified accordingly.

Minor Comments:

1. Figures 3c, d: please perform the appropriate statistical tests and report p-values in the graphs.

In response to the reviewer's comment, we increased the numbers of examined mice, and have described the p-values. The results are displayed as Fig. 3d and Supplementary Fig. 4f.

2. Figures 4B, 4C, and 5H reference phosphorylated Akt without specifying the site of phosphorylation. The specific phosphorylation site assessed in each experiment should be listed in the figure or figure legends (e.g. p-AKT (Ser473)). Also, for the flow cytometry of Figure 4b (p-AKT and pS6 staining), it would be preferable to use an anti- isotype antibody as the negative control.

Thank you very much for raising this point. According to the comment, we have now described specific phosphorylation sites in each experiment and added the histograms of the isotype control to Fig. 4b and 4c. We also confirmed the specificity of antibody against p-Akt (S473) and p-S6 by isotype control antibody (negative control) and SCF stimulation (positive control). The results are now displayed as Supplementary Fig. 6b.

3. Several of the co-immunoprecipitation experiments in Figure 5 propose an interaction between Akt and ASXL1-MT without the use of a control antibody. The authors should show a

representative experiment including an IgG control to demonstrate that the co-IP is specific and does not represent the very common artefact of non-specific protein adsorption to beads.

In response to the reviewer's thoughtful comment, we have performed experiments using a control IgG and confirmed the specific binding between AKT and ASXL1. We would like to show this result to the reviewer (See attached figure). If the editor thinks it better to include this figure in the paper, we would abide by the editor's decision.

4. In Figures 5a-f, the experiments involve overexpression in 293 T cells. Can the authors also perform the experiments in bone marrow cells, using antibodies to either the endogenous proteins or to ectopically-expressed epitope-tagged proteins?

Thank you very much for the important point. According to the reviewer's comment, we tried to detect the interaction of endogenous ASXL1 and AKT as well as retrovirally transduced epitope-tagged ASXL1 and Akt in c-kit-positive bone marrow cells. However, we could not detect the interaction probably because of the fragility of ASXL1.

5. In Figure 6b, the authors should explain the meaning behind the color scale in their heatmap (Z-score?).

Yes, the heatmap indicates Z-score. We have now described it in Supplementary Fig. 8a.

6. In Supplementary Figure 4a, the authors may want to add the frequency of the LSK population in p53 KO mice for reference.

In response to the reviewer's comment, we have now shown the frequency of LSK cells in control, ASXL1-MT/p53-WT, ASXL1-MT/p53-KO, and ASXL1-WT/p53-KO mice (Supplementary Fig. 9a).

7. RNA-seq of HSPC from young ASXL1-MT KI mice revealed upregulation of genes involved in observed mild activation of AKT/mTOR and oxidative phosphorylation. Does the RNA-seq reveal anything significant about the expression of glycolytic pathway genes in young and old mice?

Thank you very much for the important question. In response to the reviewer's comment, we performed GSEA of the RNA-seq data. We found that there was a tendency for the increased

glycolysis in HSPC of young *Vav-Cre* ASXL1-MT KI mice without a statistic significance while oxidative phosphorylation was significantly upregulated. On the other hand, glycolysis was significantly upregulated in LT-HSCs of aged *Vav-Cre* ASXL1-MT KI mice while oxidative phosphorylation was not upregulated. However, the levels of mitochondrial membrane potential and ROS were clearly increased in aged *Vav-Cre* ASXL1-MT KI mice, suggesting activation of aerobic respiration. The reason for the discrepancy in the RNA-seq analyses between young and aged *Vav-Cre* ASXL1-MT KI mice is not clear. According to the analysis of public data (GSE48893), genes related to oxidative phosphorylation is paradoxically downregulated in aged wild-type mice compared with young wild-type mice while aerobic respiration is increased with age (Theodore et al, *Nature*, 2017), implying that mitochondrial activity is not correlated with gene expression in aged mice. Although we could add these results with discussion described above, to make the paper straightforward and avoid any confusion of the readers, we would like not to present this result if the reviewer and the editor would agree. We would abide by the editor's decision.

8. In figure 7, the authors detected mitochondrial membrane potential and ROS after Rapamycin treatment, is it possible to detect the DNA damage, for example gammaH2A.X staining.

Thank you very much for the important comment. In response to the reviewer's comment, we examined the effects of Rapamycin treatment on DNA damage during competitive transplantation of aged *Vav-Cre* ASXL1-MT KI bone marrow cells. We confirmed that an increase in DNA damage (estimated by γ -H2AX) as well as mitochondrial activation by ASXL1-MT in aged mice was suppressed by Rapamycin treatment, suggesting that activation of Akt/mTOR pathway is responsible for increased DNA damage in aged ASXL1-MT KI mice. These results are now displayed in Fig. 9f-j, and the text has been modified accordingly.

References

1. Scheuermann JC, de Ayala Alonso AG, Oktaba K, Ly-Hartig N, McGinty RK, et al. Histone H2A deubiquitinase activity of the Polycomb repressive complex PR-DUB. *Nature*. 2010;465(7295):243–247.
2. Yang W-L, Wang J, Chan C-H, Lee S-W, Campos AD, et al. The E3 Ligase TRAF6 Regulates Akt Ubiquitination and Activation. *Science* (80-.). 2009;325(5944):1134–1138.
3. Fan C-D, Lum MA, Xu C, Black JD, Wang X. Ubiquitin-dependent regulation of phospho-AKT dynamics by the ubiquitin E3 ligase, NEDD4-1, in the insulin-like growth factor-1 response. *J. Biol. Chem*. 2013;288(3):1674–84.
4. Suizu F, Hiramuki Y, Okumura F, Matsuda M, Okumura AJ, et al. The E3 Ligase TTC3

Facilitates Ubiquitination and Degradation of Phosphorylated Akt. *Dev. Cell.* 2009;17(6):800–810.

5. Yang W-L, Wu C-Y, Wu J, Lin H-K. Regulation of Akt signaling activation by ubiquitination. *Cell Cycle.* 2010;9(3):487–97.

6. Jensen DE, Proctor M, Marquis ST, Gardner HP, Ha SI, et al. BAP1: A novel ubiquitin hydrolase which binds to the BRCA1 RING finger and enhances BRCA1-mediated cell growth suppression. *Oncogene.* 1998;16(9):1097–1112.

7. Ventii KH, Devi NS, Friedrich KL, Chernova TA, Tighiouart M, et al. BRCA1-associated protein-1 is a tumor suppressor that requires deubiquitinating activity and nuclear localization. *Cancer Res.* 2008;68(17):6953–6962.

8. Ebner M, Lučić I, Leonard TA, Yudushkin I. PI(3,4,5)P3 Engagement Restricts Akt Activity to Cellular Membranes. *Mol. Cell.* 2017;65(3):416–431.e6.

As the reviewer kindly provided above references, we have referred some of them in the revised version.

Reviewer #2 (Remarks to the Author):

Age-related clonal hematopoiesis (ARCH) occurs when a single mutant hematopoietic stem cell (HSC) contributes to a significant clonal proportion of mature blood lineages. ARCH is associated with increased risks of de novo and therapy-related hematological neoplasms, suggesting that mutations identified in ARCH likely drive disease development. Mutations most commonly detected in CH include the epigenetic regulators DNMT3A, TET2, and ASXL1. While loss of DNMT3A or TET2 enhances HSC self-renewal, loss of ASXL1 and expression of mutant ASXL1 decrease HSC functions. The mechanisms by which mutant ASXL1 drives clonal hematopoiesis are largely unknown.

The authors examined the influence of mutant ASXL1 on physiological HSC aging using the knockin (KI) mice expressing a C-terminally truncated form of ASXL1-mutant in the hematopoietic system (ASXL1-MT-VavCre⁺ mice). They found that aged ASXL1 mutant HSCs show decreased repopulating potential in transplantation assays. RNA-seq assays revealed that Akt/mTOR pathway is activated in young ASXL1 mutant HSCs. They further showed that mutant ASXL1 interacts with BAP, leading to deubiquitination and activation of AKT. Interestingly, pharmacologically inhibition of the mTOR pathway ameliorates aberrant expansion of the HSC compartment as well as dysregulated hematopoiesis in aged ASXL1-MT-KI mice. Thus, the

authors demonstrated that mutant ASXL1 promotes HSC aging through activating the Akt/mTOR pathway.

In addition to HSC aging, the authors utilized ASXL1-MT-Mx1Cre⁺ mice to investigate the impact of mutant ASXL1 on clonal hematopoiesis. They treated ASXL1-MT-Mx1Cre⁺ mice with pI:pC to partially induce mutant ASXL1 expression in hematopoietic system and found that HSCs expressing mutant ASXL1 acquire clonal advantage during aging.

Overall, this is an interesting study that provides a novel mechanism by which mutant ASXL1 promotes HAS aging. However, studies on CH were descriptive and data were not convincing. The mechanistic link between the impact of mutant ASXL1 on HSC aging and CH is not clear. A significant amount of experiments will be needed to strengthen the manuscript and support conclusion drawn.

Thank you very much for your valuable comments. We have further improved our manuscript by performing several experiments according to your comments.

Major Criticisms/Suggestions:

1. Both young and aged mutant HSCs show decreased repopulating potential in transplantation assays as shown in Figures 1 and 2, which may be due to impaired HSC homing capability. Therefore, authors should perform homing assays using both young and aged mutants HSCs.

Thank you very much for this important comment. According to the reviewer's comment, we have performed homing assay using LSK cells of young/aged control and *Vav-Cre* ASXL1-MT KI mice, and found that LSK cells derived from young ASXL1-MT KI mice hold comparable homing when compared with young control mice. On the other hand, homing activities of LSK cells significantly decreased in aged *Vav-Cre* ASXL1-MT KI mice when compared with young and aged control mice as well as young *Vav-Cre* ASXL1-MT KI mice. These results together suggested that the decreased repopulating potential of HSCs of ASXL1-MT KI mice was not simply caused by the homing defects of HSCs, but that the homing potential is also affected in aged ASXL1-MT KI mice. We have now added new results as Supplementary Fig. 3c and 3d, and modified the text accordingly.

2. Given that the paper focuses on determining the impact of mutant ASXL1 on HSCs during aging, RNA-seq studies using aged HSCs are needed to elucidate the molecular mechanisms by which mutant ASXL1 promotes HSC aging.

Thank you very much for raising the important issue. In this study, we observed enhancement of age-associated phenotypes in hematopoiesis including anemia, myeloid-biased differentiation, hypocellular bone marrow, and expansion of surface marker-defined LT-HSCs in aged ASXL1-MT KI mice. These phenotypes were partly rescued by inhibition of Akt/mTOR pathway (Fig. 5c and 9d), suggesting that ASXL1-MT-induced activation of Akt/mTOR pathway is involved in enhanced aging of the hematopoietic system. According to the reviewer's advice, we performed RNA-seq analysis using aged LT-HSCs to elucidate the mechanisms by which ASXL1-MT accelerates aging of HSCs. This analysis suggested activation of Akt/mTOR pathway in aged ASXL1-MT KI mice as well as young ASXL1-MT KI mice. Moreover, ASXL1-MT promotes gene expression of age-associated patterns. These results together with the experiments using Akt and mTOR inhibitors indicate the involvement of ASXL1-MT-induced activation of Akt/mTOR in enhanced aging of HSCs. The results of RNA-seq analyses of aged LT-HSCs are displayed as Fig. 10 and are described in the "Results" and "Discussions" sessions.

3. Figure 4b showed that young ASXL1 mutant HSPCs show increased levels of pAKT and pS6. What about the levels of pAKT and pS6 in aged mutant HSPCs?

Thank you very much for your thoughtful comment. In response to a reviewer's comment, we examined levels of pAkt and pS6 in HSCs of aged mice. These experiments confirmed that they were increased in aged *Vav-Cre* ASXL1-MT KI mice compared to aged control mice. We have now shown these results as Fig. 4d and 4e, and modified the text accordingly.

4. Hematopoiesis varies between mice, especially during aging. However, all data from Figure 3 were generated from only two aged ASXL1-MT-Mx1Cre+ mice (n=2). Given that the conclusion of the paper is mainly based on data from Figure 3, more aged mutant mice, at least 8 to 10 mice, are needed to generate convincing and solid data.

Thank you very much for raising this important point. We agree with the reviewer that this is an important part of this paper. When we sent this paper, we could not secure enough numbers of aged *Mx1-Cre* ASXL1-MT KI mice. However, we now have enough number of them. Therefore, according to the reviewer's request, we have examined additional 20 aged mice, and obtained similar results with those of the initial two aged mice; we demonstrate that these mice exhibit anemia, thrombocytosis, myeloid-biased differentiation, mitochondrial activation, increased ROS levels, and progression of cell cycle similar to findings of *Vav-Cre* ASXL1-MT KI mice. These results have been added to Fig. 3, Supplementary Fig. 4a-d, Supplementary Fig. 11 a-c, and the text has been modified accordingly.

5. Figure 3A. The authors treated ASXL1-MT-Mx1Cre⁺ mice with pI:pC (250ug, IP) every other day for three times to induce mutant ASXL1 expression in hematopoietic cells. This treatment should induce GFP expression in majority of hematopoietic cells. What are the percentage of GFP⁺ cells following pIpC treatment?

Thank you very much for your comment. At first, the initial protocol was appropriate for partial induction of ASXL1-MT. However, as indicated by the reviewer, the same protocol induced expression of ASXL1-MT in most hematopoietic cells in peripheral blood and bone marrow in the following additional experiments using a different lot of pIpC. Therefore, we evaluated an appropriate dose of pIpC to partially induce expression of ASXL1-MT. In the revised manuscript, we used 10 ug pIpC five times every other day. In this condition, the frequency of GFP-positive cells in peripheral blood four weeks after pIpC injections was about 20%. To make the point clearer, we have now shown a time course of the percentage of GFP-positive cells in Fig. 3b.

6. pI:pC treatment mimics interferon response. Hematopoietic aging is manifested by chronic inflammation. There are increased levels of pro-inflammatory cytokines such as interferon γ (IFN γ) in both peripheral blood serum and bone marrow fluid of aged mice. The authors found that mutant HSCs (GFP⁺) expand in ASXL1-MT-Mx1Cre⁺ mice with age (Figure 3A). The increase in GFP⁺ cells with age may not be due to increased clonal hematopoiesis but increased secretion of IFN γ in mutant ASXL1 mice, inducing more mutant ASXL1 expression in hematopoietic cells. Thus, the authors should examine the levels of IFN γ and other pro-inflammatory cytokines in PB serum and bone marrow fluid of aged WT and ASXL1-MT-Mx1Cre⁺ mice with age

Thank you very much for this thoughtful comment. According to the reviewer, we measured the levels of IFN- γ and TNF- α in PB serum and bone marrow fluid of aged *Mx1-Cre* ASXL1-MT KI mice and aged control mice. However, we did not detect significant difference between them, suggesting that an increase in GFP-positive cells was not due to elevated secretion of pro-inflammatory cytokines. The results are now displayed in Supplementary Fig. 4e and 4f. and described in the manuscript.

Reviewer #3 (Remarks to the Author):

Comments from the referee:

In the first part of the present study, the authors reproduced previous published results by comparing the hematopoiesis of young and old mice in figures 1 and 2 (increased HSC frequency, decrease competitive engraftment, increased cell cycle and apoptosis) and show evidence that HSC-CD41⁺ are increased but not functional. In contrast, after crossbreeding their KI mice with Mx1-cre mice, they show that the expression of ASXL1-mut in native hematopoiesis leads, this time, to increased clonal advantage.

In the second part of the study, they investigate the mechanism by which ASXL1-mut induces growth advantage of pLT-HSC but alters their function. This is the most interesting and novel part. Using biochemical and genetic studies, they found that ASXL1-mut/BAP1 promotes AKT deubiquitination and increases its phosphorylation. This leads to an increase in mitochondrial activity mainly through mTOR, in ROS levels and in DNA damages in a P53-dependent manner. Manipulations of this pathway (ROS inhibition with NAC, overexpressing catalase or AKT/mTOR inhibition) rescued ASXL1-mut clonal disadvantage in treated mice. These experiments indicate that the AKT/mTOR pathway is involved in the cell cycle of HSC but also alters their DNA and triggers apoptosis resulting in an overall increased pool of non-functional HSC.

Finally, they checked AKT and mTOR signaling in public transcriptome of MDS patients and found enrichment suggesting that it is relevant for the pathology, making this pathway a good target to inhibit in patients with ASXL1-mut.

This is a very interesting study that helps to better understand the development of hematological malignancies but there are several aspects that need improvement or clarification.

We thank the reviewer for the kind compliment that "This is a very interesting study that helps to better understand the development of hematological malignancies".

1) In contrast to study performed with Vav-Cre mice, authors show that the expression of ASXL1-mut phenotype in native hematopoiesis, after crossbreeding with Mx1-cre mice, leads to an increased clonal advantage. This difference is explained by a deleterious effect of transplantation in ASXL1-mut-mediated clonal advantage. I found these experiments to be a bit confusing for the understanding of the paper since all the next experiments are performed with vav-Cre. Moreover, since Cre, under Mx1 promoter control, can be expressed in non-hematopoietic cells, another explanation could be that the other cells or microenvironment may play a role in this difference. What happens in term of clonal advantage in the PB if BM cells from Mx1-Cre/ASXL1-mut mice are engrafted in WT mice or conversely if Vav-Cre/ASXL1-mut BM cells are engrafted in Mx1-Cre/ASXL1-mut after induction? Alternatively, ASXL1-mut-induced

competitive advantage could be due to aging of microenvironment in addition to aging of hematopoiesis. In this regard, it would be informative to perform transplantation of *Vav-Cre/ASXL1-mut* BM cells in aged recipients to assay clonal advantage.

Thank you very much for the important comment. In response to the reviewer's comment, we performed transplantation assays with three different settings. First, we transplanted bone marrow cells from young *Mx1-Cre ASXL1-MT* KI mice into wild-type recipient mice and induced ASXL1-MT expression by pIpC injections one month after transplantation. Induction of ASXL1-MT after engraftment did not lower chimerism of peripheral blood but rather tended to increase the frequency of donor-derived bone marrow cells, suggesting that ASXL1-MT exerts competitive disadvantage on LT-HSCs, specifically during recovery from transplantation. Second, we transplanted bone marrow cells from young *Vav-Cre ASXL1-MT* KI mice into young or aged wild-type recipient mice. The frequencies of ASXL1-MT-expressing cells were similar regardless of the age of recipient mice. Finally, we transplanted bone marrow cells from young *Vav-Cre ASXL1-MT* KI mice into *Mx1-Cre ASXL1-MT* KI mice that had been induced expression of ASXL1-MT by pIpC injections one month before transplantation. Expression of ASXL1-MT in non-hematopoietic cells of the recipient bone marrow did not affect the chimerism of ASXL1-MT-expressing cells in peripheral blood. Taken together, we concluded that expression of ASXL1-MT in LT-HSCs promotes the age-related clonal expansion independent of either aging of microenvironment or ASXL1-MT-expressing stromal cells in native hematopoiesis. We have added these results as supplementary figure 5a-g, and have modified the text accordingly.

2) In their study, the authors check the mechanism underlying the growth advantage of pLT-HSC expressing ASXL1-mut and leading to their clonal disadvantage. Thus, the title seems overstated since the *mut-ASXL1* did not induce "clonal hematopoiesis" in the *Vav-Cre* mouse model.

Thank you very much for the valuable advice. In response to the previous comment, we could secure the number of aged *Mx1-Cre ASXL1-MT* KI mice that had been partially induced expression of ASXL1-MT at 12 weeks after birth. In these mice, ASXL1-MT-expressing cells gradually expanded with age, eventually replacing HSC compartment with ASXL1-MT-expressing cells. Considering that ASXL1-mutated hematopoietic cells acquire a growth advantage to expand their clone in the process of clonal hematopoiesis, we would like to argue that our findings using *Mx1-Cre ASXL1-MT* KI mice recapitulate clonal hematopoiesis harboring *ASXL1* mutations. However, as the reviewer pointed out, the title may be overstated because further evidences are required to clarify the exact mechanism for ASXL1-MT-induced clonal hematopoiesis. Therefore, we would like to change the title from "Mutant ASXL1 induces clonal

hematopoiesis through activation of Akt/mTOR pathway” to “Mutant ASXL1 induces expansion of the hematopoietic stem cell compartment through activation of Akt/mTOR pathway”.

3) The finding that AKT is overactivated in LSK is very interesting.

Are the results of flow cytometry obtained in steady state? Since sometimes the phosphorylated form of intracellular protein are difficult to detect, the authors should show the histogram plots for Ig control antibodies and, if possible, a positive control (in vitro treatment with SCF) and/or a negative control (in vitro treatment with PI3K inhibitor).

Thank you very much for raising this important point. Yes, the flow cytometry analyses were performed in the steady state cells. According to the reviewer’s suggestions, we have now shown positive (SCF-stimulated) and negative (isotype) controls in the new Supplementary Fig. 6b. We have modified the text and figure legend accordingly.

4) The demonstration that ASXL1-mut/BAP1 promotes AKT deubiquitination is novel and of great interest. Why did the authors choose to focus on AKT2 and not AKT1? Are the findings also true with AKT1?

Thank you very much for this important point. The reason why we used AKT2 mutants here was simply because only AKT2 AA and DD mutants were available. In this experiment, we tried to confirm the published result that ASXL1 preferentially binds phosphorylated AKT (Ref. 44). In response to the reviewer’s comment, we have now performed the experiment using AKT1 AA and DD mutants. However, we could not obtain the similar results with those obtained by using AKT2 mutants. Although the reason for this discrepancy remains elusive, this does not affect the conclusion of the paper. Therefore, we would like to delete this non-essential result from the original paper (original Fig. 5c and 5d), which will not affect the conclusion of the paper, to avoid any confusion of the reader.

5) Moreover, since the ASXL1-mut/BAP1 promotes AKT deubiquitination and stabilization, is total AKT increased in LSK? In 293T cells, is AKT half-life affected by ASXL1-mut? Maybe cycloheximide experiments should be conducted?

Thank you very much for raising this point. In response to the reviewer’s comment, we performed intracellular flow cytometry analyses. The expression level of total Akt was not increased in LSK cells of *Vav-Cre* ASXL1-MT KI mice (Supplementary Fig. 6a). As phosphorylated form of Akt is exclusively ubiquitinated and degraded (Ref. 38, 40-42), the expression level of total Akt

should not be significantly altered. Therefore, we did not perform cycloheximide experiments. The result has now shown in Supplementary Fig.6a and described in the manuscript.

6) The experiment performed on P53^{-/-}-ASXL1-mut are interesting to understand the mechanism of action. Did the authors check the HSC compartment in these mice? Since p53 impacts genetic instability and ROS level through transcriptional mechanism, it could be informative to measure ROS levels and DNA damages in these mice.

Thank you very much for the reviewer's comment. According to the reviewer's comment, we examined LT-HSC compartment in P53^{-/-}-ASXL1-mut mice. Similarly to LSK cells, the frequency of LT-HSCs was significantly increased compared to that of ASXL1-MT KI mice. The result is now displayed as Supplementary Fig. 9b.

Since we could not secure the number of p53-KO/ASXL1-MT KI mice, we expressed a dominant-negative form of p53, p53DD, in ASXL1-MT KI BM cells to answer the question of the reviewer. We found the increased levels of ROS and DNA damage in these cells, suggesting an important role of p53 in the regulation of ROS levels and DNA damage response in ASXL1-MT KI mice. We have added these results as Supplementary Fig. 9g-j, and modified the text accordingly.

Minor

Authors should verify the indicated number of mice in figure legends. For instance, in figure 1 (h) there seem to be 5 mice in the figure and not 6.

In addition, authors should detail in figure legends the mouse model that was used for each experiment: Vav-Cre or Mx1-Cre.

Thank you very much for the comment. We have corrected the number of the mice and have added more details in the figure legend.

FLAG - ASXL1	-	-	WT	WT	WT	MT	MT	MT
AKT1	-	-	WT	AA	DD	WT	AA	DD
BAP1	-	+	+	+	+	+	+	+

REVIEWERS' COMMENTS

Reviewer #1 (Remarks to the Author):

The authors have done a tremendous amount of work to revise the manuscript and the revised manuscript is now suitable for publication.

I did find a few typos and mistakes:

the legend to Supplementary Figure 2d appears to be missing

there is a typo in Figure 3a: complete "bloot" count -- should be "blood"

the legend to Supplementary Figure 4 should be checked -- according to the figure itself and the figure description in the text, the cytokine data should be in e and f

there is a typo in Supplementary Figure 11d: "perisosine" should be "perifosine"

Reviewer #2 (Remarks to the Author):

The authors made an significant effort in addressing reviewer's comments. The manuscript is much improved.

Reviewer #3 (Remarks to the Author):

The revised manuscript provided by the authors is much clearer and robust than the initial version. It was nice that the authors added the flowcharts in the figures because it leads to an improvement of the understanding.

I think that the authors have answered all the concerns I raised and I particularly appreciated they have very well investigated all the phenotypes in young and aged mice and have tested the cell-intrinsic and cell extrinsic mechanisms that mediate the effect of ASXL1-MT on HSC.

I have just two remarks:

Flow chart fig 3: typo in "bloot" instead of "blood"

Concerning the AKT1 experiments, it should be good to put the vectors used in material and methods

Reviewers' comments:

Reviewer #1 (Remarks to the Author):

The authors have done a tremendous amount of work to revise the manuscript and the revised manuscript is now suitable for publication.

I did find a few typos and mistakes:

the legend to Supplementary Figure 2d appears to be missing

there is a typo in Figure 3a: complete "bloot" count -- should be "blood"

the legend to Supplementary Figure 4 should be checked -- according to the figure itself and the figure description in the text, the cytokine data should be in e and f

there is a typo in Supplementary Figure 11d: "perisosine" should be "perifosine"

We have revised the manuscript according to the reviewer's comments. We appreciate your thoughtful suggestions.

Reviewer #2 (Remarks to the Author):

The authors made an significant effort in addressing reviewer's comments. The manuscript is much improved.

Thanks to your comments, our paper has been much improved.

Reviewer #3 (Remarks to the Author):

The revised manuscript provided by the authors is much clearer and robust than the initial version. It was nice that the authors added the flowcharts in the figures because it leads to an improvement of the understanding.

I think that the authors have answered all the concerns I raised and I particularly appreciated they have very well investigated all the phenotypes in young and aged mice and have tested the cell-intrinsic and cell extrinsic mechanisms that mediate the effect of ASXL1-MT on HSC.

I have just two remarks:

Flow chart fig 3: typo in “bloot” instead of “blood”

Concerning the AKT1 experiments, it should be good to put the vectors used in material and methods

We have modified the manuscript according to the reviewer’s comments. We could improve our research with all your help.